# 21st Century drought-related fires counteract the decline of Amazon deforestation carbon emissions

Luiz E.O.C. Aragão [1,2], Liana O. Anderson[3,4], Marisa G. Fonseca[1], Thais M. Rosan[1], Laura B. Vedovato[1], Fabien H. Wagner[1], Camila V.J. Silva[1,5], Celso H.L. Silva Junior [1], Egidio Arai[1], Ana P. Aguiar[6,7], Jos Barlow [5], Erika Berenguer[4,5], Merritt N. Deeter[8], Lucas G. Domingues[6,9], Luciana Gatti[6,9], Manuel Gloor[10], Yadvinder Malhi [4], Jose A. Marengo[3], John B. Miller[11], Oliver L. Phillips[10] & Sassan Saatchi[12]

Tropical carbon emissions are largely derived from direct forest clearing processes. Yet, emissions from drought-induced forest fires are, usually, not included in national-level carbon emission inventories. Here we examine Brazilian Amazon drought impacts on fire incidence and associated forest fire carbon emissions over the period 2003–2015. We show that despite a 76% decline in deforestation rates over the past 13 years, fire incidence increased by 36% during the 2015 drought compared to the preceding 12 years. The 2015 drought had the largest ever ratio of active fire counts to deforestation, with active fires occurring over an area of 799,293 km$^2$. Gross emissions from forest fires (989 $\pm$ 504 Tg CO$_2$ year$^{-1}$) alone are more than half as great as those from old-growth forest deforestation during drought years. We conclude that carbon emission inventories intended for accounting and developing policies need to take account of substantial forest fire emissions not associated to the deforestation process.

[1] Remote Sensing Division, National Institute for Space Research, Av. dos Astronautas, 1.758, 12227-010 São José dos Campos, Brazil. [2] College of Life and Environmental Sciences, University of Exeter, Exeter EX4 4RJ, UK. [3] National Centre for Monitoring and Early Warning of Natural Disasters - Cemaden (CEMADEN), São Jose dos Campos, Brazil. [4] School of Geography and the Environment, University of Oxford, Oxford, OX1 3QY, UK. [5] Lancaster Environment Centre, Lancaster University, Lancaster LA1 4YQ, UK. [6] Earth Systems Sciences Center, National Institute for Space Research, Av. dos Astronautas, 1.758, 12227-010 São José dos Campos, Brazil. [7] Stockholm Resilience Centre, Stockholm University, Kräftriket 2B, Stockholm SE-10691, Sweden. [8] National Center for Atmospheric Research, Atmospheric Chemistry and Observations Laboratory, Boulder, CO 80301, USA. [9] Instituto de Pesquisas Energéticas e Nucleares (IPEN)–Comissao Nacional de Energia Nuclear (CNEN)–Atmospheric Chemistry Laboratory, 2242 Avenida Professor Lineu Prestes, Cidade Universitaria, Sao Paulo CEP 05508-000, Brazil. [10] School of Geography, University of Leeds, Leeds LS2 9JT, UK. [11] NOAA/Earth System Research Laboratory/Global Monitoring Division, and Cooperative Institute for Research in Environmental Science (CIRES), University of Colorado Boulder, Boulder, CO 80305, USA. [12] Jet Propulsion Laboratory, California Institute of Technology, 4800 Oak Grove Drive, Pasadena, CA 91109, USA. Correspondence and requests for materials should be addressed to L.E.O.C.Aão. (email: luiz.aragao@inpe.br)

Land use and land cover changes (LULCC) in tropical countries contribute significantly to greenhouse gas emissions and play a major role in changing the global climate[1, 2]. Curbing deforestation, defined here as the complete conversion of old-growth forests into productive lands, is expected to directly reduce tropical carbon (C) emissions[3, 4]. In the Brazilian Amazon, annual deforestation rates have been closely linked ($r^2 = 84\%$, $p < 0.004$) to annual fire incidence[5], with fires being the main pathway for removing plant biomass and transferring the associated C from the tropical vegetation to the atmosphere[6]. By 2015, Brazil had accomplished a 66% reduction in Amazonian deforestation rates (6207 $km^2$ $year^{-1}$) when compared to the 1988–2004 mean (18,439 ± 5121 $km^2$ $year^{-1}$)[7]. This achievement followed a set of civil society, economic and public policy actions, most notably the Action Plan for Prevention and Control of Deforestation in Amazonia (PPCDAm), implemented in 2004 by the Brazilian government with three phases (2004–2008, 2009–2011 and 2012–2015) aiming to continuously reduce illegal deforestation and establish a sustainable development model for Amazonia[8].

While emissions due to deforestation have fallen, it has been shown that during drought years, fire incidence and associated C emissions in human-modified regions have increased[6, 9]. In 2010, a drought year, gross C emissions due to fires were 1.7 times higher (0.51 ± 0.12 Pg C $year^{-1}$) than during the subsequent non-drought year[6]. This corresponded to 57% of 2010 global emissions from land use change (0.9 ± 0.7 Pg C)[10]. Most Earth System Models (ESMs) predict increasing dry season intensity in Amazonia in the 21st century. This is directly related to radiative forcing[11] and declining Northern Hemisphere aerosol production[12, 13], which tend to cause anomalous variation of sea surface temperatures (SST) and consequently drive large-scale swings in precipitation over Amazonia. If this new climatic configuration is borne out, Amazonia is expected to become an amplified fire-prone system[14, 15], with emissions from drought-induced fires unrelated to deforestation increasingly playing a much larger role than those from deforestation[15].

We hypothesize that if the influence of drought on fire incidence prevails over that of deforestation, then the reduction in deforestation rates by 2015 would not lead to a direct reduction of C emissions. Specifically, we predict that drought will combine with human activities other than deforestation, including secondary vegetation slash-and-burn and cyclical fire-based pasture cleaning. These alone provide sufficient ignition sources for fire to leak into adjacent forests—many of which are fragmented or degraded and therefore more likely to burn[16–18]. Recent drought events (i.e., 2005, 2010 and 2015) can serve as a model for assessing how oceanic modes shift the amount and distribution of Amazonian rainfall, in turn affecting spatio-temporal patterns of fire-prone regions in future climate conditions. These events, therefore, provide a unique opportunity to quantify the sensitivity of drought-fire interaction under the current trend of reduced deforestation rates in the Brazilian Amazon.

To test the hypothesis, we assess the causes of recent Brazilian Amazon droughts and quantify their impact on forest fire-associated C emissions patterns under the current deforestation reduction trajectory. We analyze 13 years of monthly time series (2003–2015) of sea surface temperature anomalies (SSTA) indices[19–22] combined with a suite of satellite-derived rainfall[23], active fire detections[24], atmospheric carbon monoxide (CO)[25], annual deforestation[7] and burned area data for the Brazilian Amazon.

We show that C emissions from Amazonia are increasingly dominated by forest fires during extreme droughts, rather than the prevalence of emissions from fires directly associated with the deforestation process. Forest fires alone are currently contributing to a mean annual committed gross emission of 454 ± 496 Tg $CO_2$ $year^{-1}$ (2003–2015) or 31 ± 21% of the estimated emission from deforestation. We conclude that the Brazilian Amazon may be entering a new land use and land cover change phase in which a decoupling between fire-related and deforestation-related carbon emissions, driven by recurrent 21st century droughts, can undermine the Brazilian achievement of reducing emissions from deforestation. Therefore, policy actions must amplify the focus on new practices of land management prioritizing a reduction of non-deforestation fire-use and more careful fire management, while restraining deforestation rates.

## Results

**Spatio-temporal coherence between active fires and droughts.** Our analysis of SSTA showed that the two strongest droughts in this century, occurring in 2005 and 2010, were strongly correlated with the anomalous warming of the Atlantic ocean captured by the Atlantic Multidecadal Oscillation (AMO)[19, 20] index (Fig. 1a–e). In contrast, the 2015 drought occurred following a simultaneous development of anomalous warming of the equatorial and eastern tropical north Pacific and tropical north Atlantic oceans, as measured by the Multivariate El Nino Index (MEI)[21], the Pacific Decadal Oscillation (PDO)[22] and the AMO[19, 20] indices, respectively (Figs 1a–e and 2a). The positive MEI, PDO and AMO consistently matched the intensification of negative rainfall anomalies (see Methods section for details) up to four standard deviations ($\sigma$, $p < 0.01$) towards the end of 2015 (Fig. 1d). This rainfall shortage caused the largest basin-wide mean water deficit (–95 mm $month^{-1}$) observed since 2003 (Fig. 1e), creating a widespread drying condition that escalated active fire occurrence over the Brazilian Amazon. Active fire anomalies reached over $2\sigma$ ($p < 0.05$, Fig. 1f), surpassing fire events of previous 2005 and 2010 drought years towards the end of 2015 (Supplementary Fig. 1).

By performing a pixel-based correlation between gridded rainfall anomaly and oceanic indices, we showed that the influence of SSTA on rainfall reduction in Amazonia is spatially variable (Figs 1f, 2a–e). While a positive AMO negatively correlates with rainfall anomalies in south-western Brazilian Amazon (Fig. 2d), an El Niño (positive MEI) reduces rainfall in the north, central and southeast flanks of the region (Fig. 2b). The inverse correlation between PDO and rainfall mostly overlaps the grid cells influenced by MEI in the northern-central part of the region (Fig. 2c).

The effect of SSTA-driven differential rainfall shortage is evident when analyzing the spatial patterns of maximum cumulative water deficits (MCWD) and consequent enhancement of active fire incidence during major droughts (Fig. 3a–m). The MCWD anomaly pattern clearly followed the spatial configuration of the relationship between SSTA and rainfall anomalies, specifically indicating a dominant influence of the AMO during the 2005 (Fig. 3a, d) and 2010 (Fig. 3b, e) droughts and of all three indices during 2015 (Fig. 3c, f).

The 2015 drought (Fig. 3c) was the most extreme of the 21st century, as an area of 1,832,488 $km^2$ (43% of the Brazilian Amazon biome and >7 times the area of the United Kingdom) experienced significant negative MCWD anomalies (number of grid cells with s.d. ($\sigma$) larger than 1.65, $p < 0.1$), in comparison to 2005 (22%) and 2010 (25%) (Fig. 3a–f). A total of 608,764 $km^2$ experienced MCWD anomalies $>3\sigma$ ($p < 0.003$) in 2015, especially in central and eastern Amazonia, in comparison to 312,902 and 164,970 $km^2$ in 2005 and 2010, respectively (Fig. 3d–f). Most strikingly, unlike the 2005 and 2010 droughts, active fire detections associated with the 2015 drought extended beyond the Arc of Deforestation (Supplementary Fig. 2),

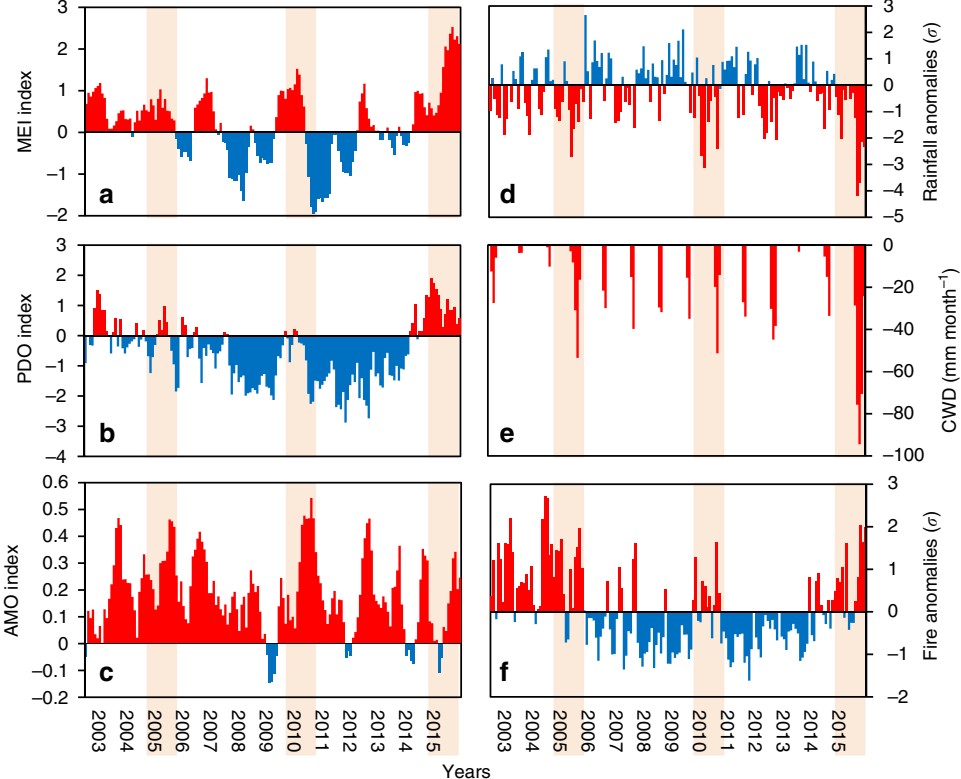

**Fig. 1** Temporal patterns of oceanic indices and their influence on droughts and fires. Monthly anomalies of sea surface temperatures (SSTA) as measured by **a** the Multivariate El Niño index (MEI), **b** the Pacific Decadal Oscillation (PDO), **c** the Atlantic Multidecadal Oscillation (AMO). Rainfall anomalies (mm month$^{-1}$) (**d**) and monthly cumulative water deficit (CWD) (mm month$^{-1}$) (**e**) were calculated based on the monthly mean of all 0.25° grid cells from the Tropical Rainfall Measuring Mission (TRMM) within the limits of the Amazon biome inside the Brazilian Amazon. Fire anomalies (s.d.) (**f**), were calculated based on the monthly sum of all active fires detected within this geographical limit. Rainfall and fire anomalies were calculated as the departure from the 2003 to 2015 monthly averages, excluding drought years (2005, 2010 and 2015). Red and blue bars indicate respectively positive and negative anomalies for all variables, except rainfall, which is displayed with reverse colours. Shaded areas indicate drought years

impacting areas in central Amazonia barely affected by fires in the past (Fig. 3g–i). In 2015, 19% (799,293 km$^2$) of the grid cells, with at least one fire detection during the studied period, experienced significant positive ($p < 0.1$) active fire anomalies. Moreover, the area affected by positive active fire anomalies surpassing $2\sigma$ ($p < 0.05$) in 2015 (628,901 km$^2$) was about double of that observed in 2005 and 2010 droughts (363,245 and 388,803 km$^2$, respectively; Fig. 3j–m).

**Impact of droughts on forest fire-associated C emissions.** While annual deforestation rates (2003–2015) had a significant declining trend of 1796 ± 271 km$^2$ yr$^{-1}$ (Fig. 4a; $R^2 = 80\%$; $p < 0.001$), the active fire counts trend was insignificant (Fig. 4b). Even with the mean deforestation rate of the third phase of PPCDAm programme (2012–2015—5420 ± 759 km$^2$) being about one-third of that recorded in the first phase (2004–2008—17,127 ± 6,571 km$^2$), the number of active fires detected by the MODIS sensor in 2015 (114,558 fires) was 15% higher than the mean number of fires recorded during the first PPCDAm phase (99,700 ± 37,940 fires). Despite 78% lower deforestation rates than in 2004 (27,772 km$^2$), 2015 had one of the longest fire seasons (5 months with over 10,000 fires detected) recorded in the 21st century (Fig. 5). Deforestation explained 84% of active fire detections[5] in the pre-PPCDAm period, but only 47% during the full 2004–2015 PPCDAm interval (Supplementary Fig. 3).

The drought-induced decoupling of deforestation and active fires were supported by our analysis of forest fire-associated C

emissions. Estimated Brazilian Amazon CO$_2$ emission levels[26] have declined significantly ($R^2 = 90\%$; $p < 0.001$) with a rate of 130.8 ± 15.24 Tg CO$_2$ year$^{-1}$ from 2003 to 2012 (Fig. 4c). Despite responding to fire peaks during recent drought years, MOPITT-derived CO concentration in the total atmospheric column, a tracer for fire emission contribution to the atmospheric carbon burden[6], over the Brazilian Amazon did not followed the reported reduction in CO$_2$ emissions, as no significant temporal trend in this data set was observed (Fig. 4d). Moreover, the estimated Brazilian CO$_2$ net emissions was strongly dependent ($R^2 = 0.99$; $p < 0.001$) on the annual area deforested, while MOPITT-derived CO concentration was not (Fig. 4e).

Further analysis reinforces these results: by calculating the number of active fires per square kilometre deforested, we found that from 2003 to 2015 an increased number of fires were detected per km$^2$ of area deforested ($R^2 = 63\%$; $p < 0.01$); similarly, the concentration of CO in the total atmospheric column per km$^2$ of area deforested ($R^2 = 86\%$; $p < 0.001$) increased for the same time period; finally, no significant temporal trend was observed for the concentration of CO in the total atmospheric column per active fire count (Supplementary Fig. 4).

By analyzing burned area data from 2008 to 2012 (Fig. 6a), we further verified, that there was no evidence of an increase in pasture area burned in comparison to forest area burned that could explain the observed weakening of the deforestation-active fire correlation (Supplementary Fig. 3) or the lack of correlation

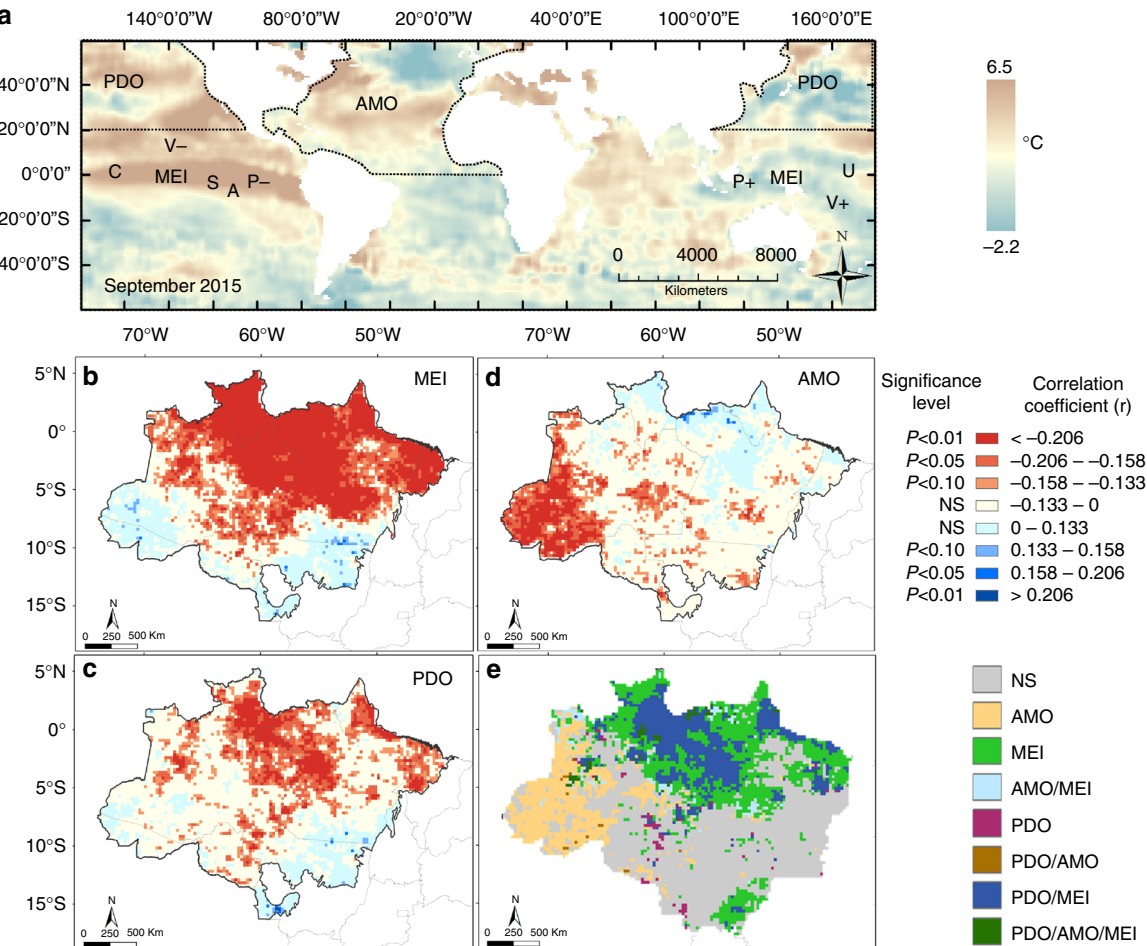

**Fig. 2** Spatial location of oceanic modes and their influence on the spatial distribution of rainfall reduction. **a** Sea surface temperature anomaly (SSTA) based on September 2015 monthly Reyn_SmithOIv2 global product (methods). Values are in degrees Celsius (°C). The figure display the regions used for calculation of the AMO[25] and PDO[27] indices as well as the approximate locations of the variables used for calculation of the MEI[26, 44]. P+ and P− represent sea-level pressure dipole, U corresponds to the zonal westerly wind anomalies in central Pacific ocean, V+ and V− correspond to the southerly and northerly meridional wind anomalies, S is the sea surface temperature and A the surface air temperature, and C indicates cloudiness anomalies. Pixel-based monthly correlations between the oceanic indices and rainfall anomalies were calculated for **b** MEI, **c** PDO and **d** AMO. Classes in the map with red colour shades indicate that the increased value of one oceanic index has the effect of reducing rainfall in the specific grid cell (negative correlation). Blue colour shades indicate the opposite. **e** Mapped grid cells with significant negative correlation coefficients ($p < 0.05$) between oceanic indices such as the MEI (**b**), PDO (**c**) and AMO (**d**) and monthly rainfall anomalies. Classes in the map indicate where the increased value of one oceanic index or their combination has the effect of reducing rainfall in the specific grid cell. All grid cells have a 0.25° spatial resolution

between MOPITT-derived CO concentration and deforestation rates (Fig. 4e). In fact, 95% ($p > 0.01$) of the variance of the MOPITT-derived CO concentration was explained by log transformed area of burned forest (Fig. 6b, d). We estimated that mean gross emissions from forest fires (old-growth plus secondary forests) alone ($454 \pm 496$ Tg $CO_2$ year$^{-1}$) from 2003 to 2015 were not statistically different (two-tailed Student's $t$-test, $t$-value $= -0.01$, degrees of freedom $= 9$) from deforestation gross emissions ($702 \pm 403$ Tg $CO_2$ year$^{-1}$) (Fig. 6e). Forest fires contributed on average with $31 \pm 21\%$ of the emission values from deforestation, with contributions beyond 50% for 2005, 2007, 2010 and 2015.

Lastly, using both active fire and burned area data, we carried out an analysis demonstrating that areas with high fire activity are not always related to high deforestation activity (Fig. 7a, b). While high levels of active fire incidence were concentrated in grid cells presenting maximum observed values of deforestation rate (excepting 2010, 2011 and 2013), following the expected pattern (Fig. 7a), positive fire anomalies have gradually increased from 11% in 2003 to 25% in 2015 in grid cells with no deforestation.

An increased trend of positive fire anomalies was also observed for grid cells with deforestation rate values below the median (1.43 km² of deforestation). Since 2009, positive fire anomalies were recurrently recorded in over 36% of grid cells with deforestation rates below 0.26 km². This pattern was neither observed during the Pre-PPCDAm period nor during the first phase of the programme. At the end of the third PPCDAm phase, corresponding to the 2015 drought, positive fire anomalies were observed in over 50% of grid cells in almost all classes of deforestation. Assessing both 2010 and 2015 droughts, we observed that all deforestation classes had more than 38% of the grid cells with positive fire anomalies, indicating an increased number of ignition sources and potential fire leakage to adjacent forest edges. The increased extent of forest fires is confirmed by the burned area map information of secondary and old-growth forests classes, showing that even in grid cells without any recorded deforestation, we still find an increase in forest area burned from 2008 (150–200 km²) to 2012 (300–350 km²; Fig. 7b). Finally, during the drought year of 2010, 90.5%

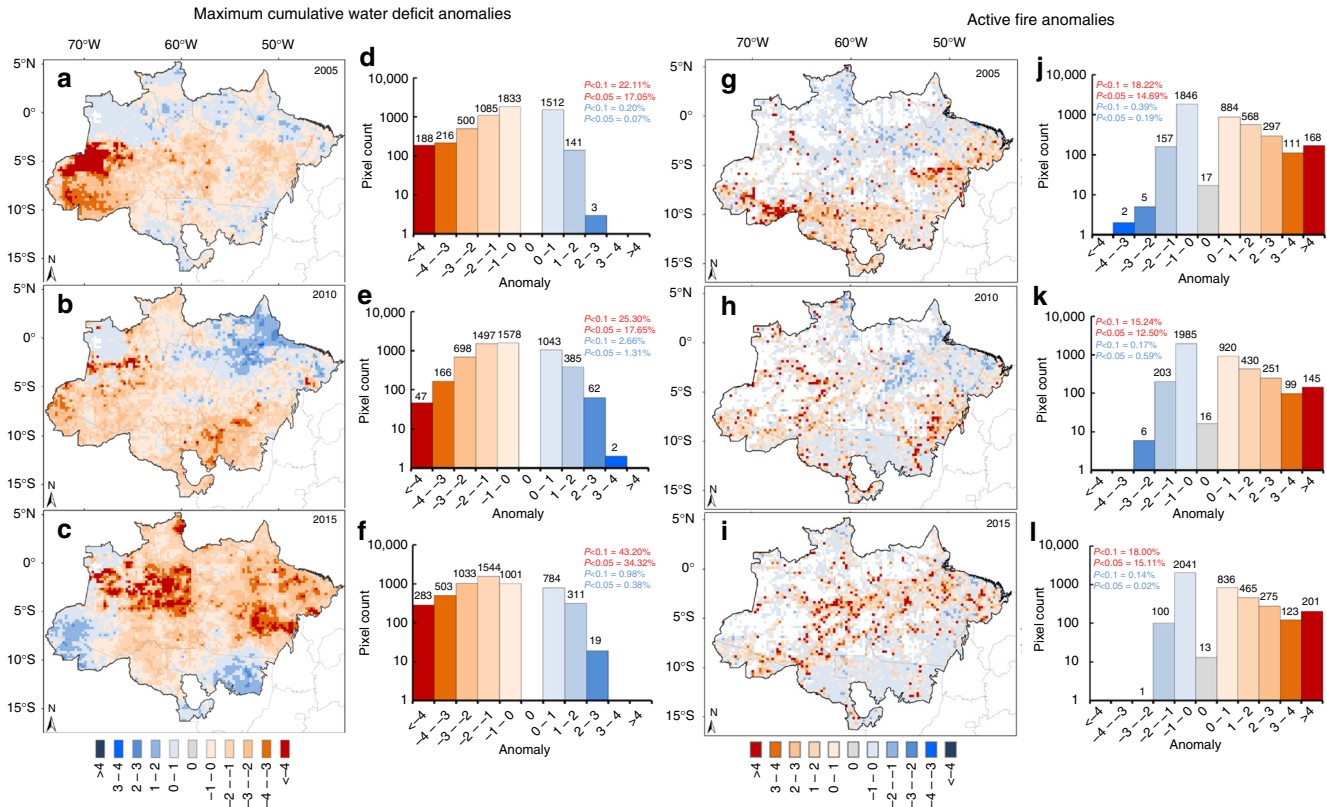

**Fig. 3** Spatial patterns of water deficit and active fire incidence in the Brazilian Amazon. Pixel-based annual anomalies ($\sigma$) of the Maximum Cumulative Water Deficit (MCWD) for 2005 (**a**)), 2010 (**b**) and 2015 (**c**) and the histograms showing the pixel counts for each category in the MCWD map for 2005 (**d**), 2010 (**e**) and 2015 (**f**). Similarly, we show maps of active fire anomalies for 2005 (**g**), 2010 (**h**) and 2015 (**i**) and their respective histograms (**j**), (**k**), (**l**). Red and blue bars indicate respectively positive and negative anomalies for fires and the opposite for the MCWD. All units are standard deviation values calculated as the departure of annual values from the 2003–2015 annual averages, excluding drought years (2005, 2010 and 2015)

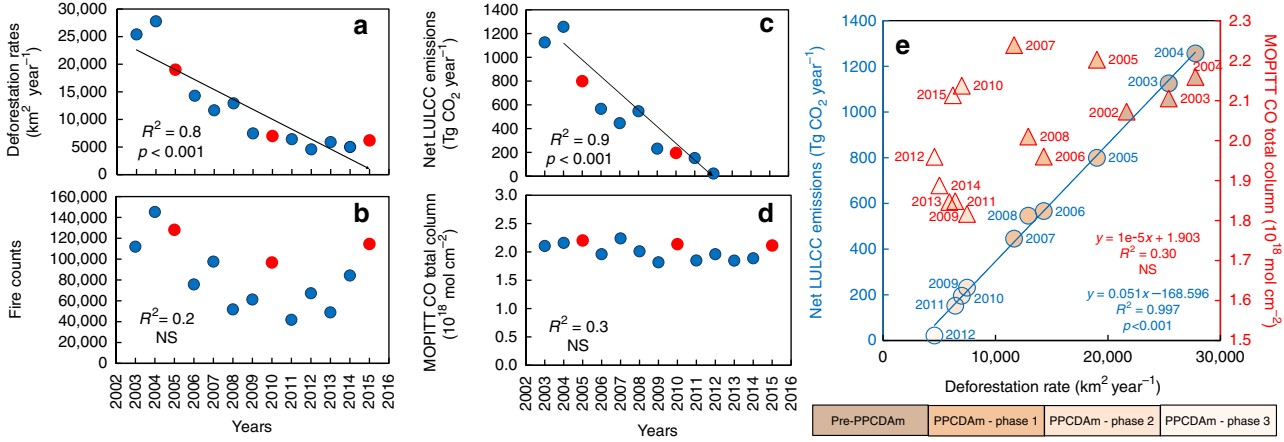

**Fig. 4** Annual trends in deforestation and forest fire-associated carbon emissions in the Brazilian Amazon. Linear trends (2003–2015) of annual **a** deforestation rates, **b** active fires counts, **c** reported Brazilian net land use and land cover change-related $CO_2$ emission estimates[4] (Net LULCC emissions) and **d** Measurements of Pollution in the Troposphere (MOPITT) CO total column data. Red circles indicate the analyzed drought years. Linear trends (black lines) are shown for statistically significant data. In **a–d** $R^2$ is the coefficient of determination, $p$ is the probability calculated at 95% confidence level and NS indicate non-significant trends. The relationship between the Net LULCC emissions and MOPITT CO total column with deforestation rates are shown in **e**. Symbols are grouped with different colours to separate the years according to four PPCDAm periods

of all deforestation classes had burned more than during all other years.

## Discussion

Our results corroborate previous findings demonstrating significant relationships between the AMO and the 2005 and 2010 drought-mediated rainfall shortage[27, 28], which led to consequent increases of fire incidence in Amazonia. Interestingly, the 2015 drought emerged from a more complex combination of positive anomalies in the three main oceanic modes analyzed here. The temporal and spatial pattern of rainfall anomalies in Amazonia during 2015 was a result of a simultaneous development of

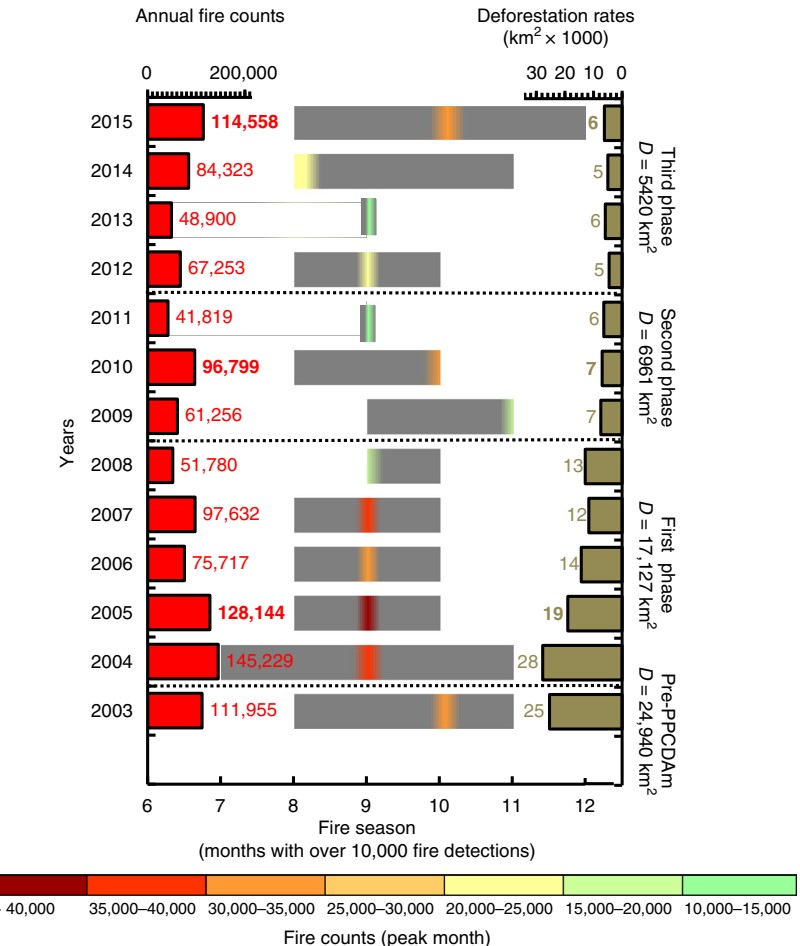

**Fig. 5** Annual fire behaviour in the Brazilian Amazon. Fire season estimated as the months with over 10,000 fire detections. Numbers in the x-axis correspond to the fire season months from June (month 6) to December (month 12). The length of the grey bars corresponds to the sum of all months with fire above the threshold. Colours inside of grey bars indicate the number of active fires detected during the peak fire month. Red bars on the left indicate the absolute number of annual fires detected and dark green bars on the right are depicting the total area deforested (km$^2$ yr$^{-1}$) for each year within the limit of the Brazilian Amazon biome. Dashed lines separate the phases of the Action Plan for Prevention and Control of Deforestation in Amazonia (PPCDAm). Numbers over the bars in bold correspond to the drought years

anomalous warming of the equatorial and eastern tropical north Pacific and tropical north Atlantic oceans, as measured by MEI[21], PDO[22] and AMO[19, 20] indices. This configuration was responsible for changes in large-scale atmospheric circulation patterns associated to the Hadley Cell and Walker Circulation[29], with subsidence over Amazonia resulting in below-average rainfall during the summer-fall season of 2015.

According to our analysis of the three major Amazonian droughts, the 2015 event was the most extreme of the 21st century[30]. Most strikingly, unlike the 2005 and 2010 droughts, fires associated with the 2015 drought extended beyond the Arc of Deforestation (Supplementary Fig. 2), impacting areas in central Amazonia barely affected by fires in the past. Our results emphasize the fact that in a hotter and drier future, large swaths of the Amazon, distant from the main deforestation epicentres, may burn.

Currently, peaks in active fire detection are more strongly related to extreme drought events than to deforestation, as used to be the case[5]. This is confirmed by the progressive temporal decoupling between fire occurrence and deforestation activities[31]. Deforestation is clearly losing its explanatory power over the variance of the absolute number of fire detections, with this

relationship gradually degrading from the first to the third PPCDAm phase (Supplementary Fig. 3).

The observed disassociation between deforestation and active fire incidence can be related to increased fire incidence in either (i) already deforested land covered by pastures (with no net impact on the atmospheric C burden) or (ii) forested areas dominated by woody vegetation. Our analysis showing that the amount of CO emitted per fire count is temporally stable indicates that it is unlikely that C emissions are primarily related to fires in pastures. This is supported by the fact that a shift in dominance from deforestation fires to pasture fires, through time, would lead to a reduction in the CO concentration per fire count. This is expected because emissions from the combustion of pasture grasses produces 10 times less C per burned area than the combustion of woody material[32].

Using an independent data set of burned area mapped during 2008–2012, we further show that 95% of the variance in the MOPITT-derived CO concentration was explained by the total area of understory old-growth plus secondary forest fires ($F = 56.4$, $p = 0.005$), while a non-significant relationship was found between MOPITT-derived CO concentration and PRODES-derived deforested area. These results support our expectation

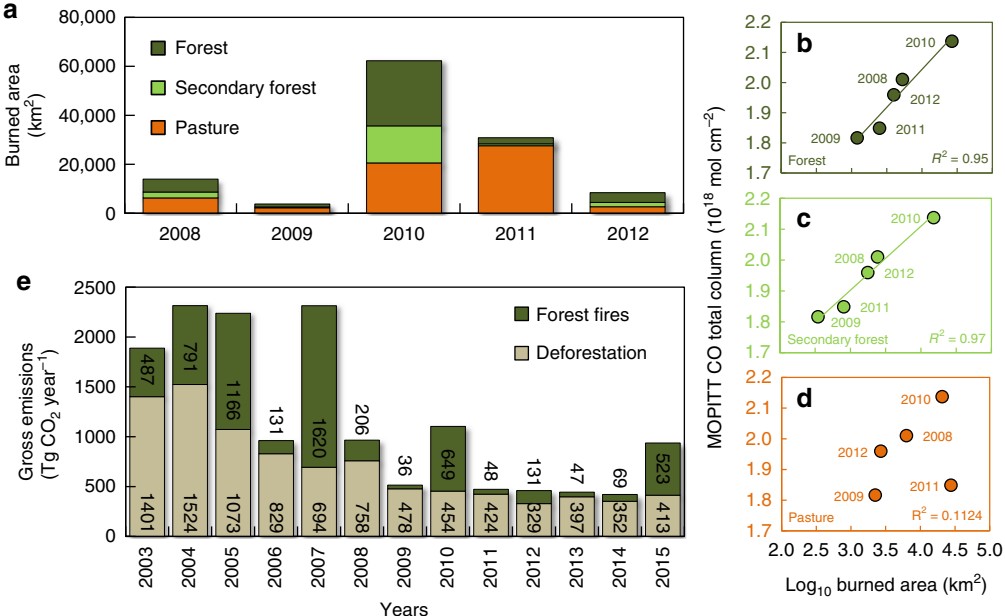

**Fig. 6** Burned area by land cover and forest fire-associated gross C emissions in the Brazilian Amazon. **a** Annual area burned quantified from the classification of MODIS images from 2008 and 2012 for three land cover types: forest, secondary forest and pasture. Panels on the right indicate the relationship between log transformed annual area burned in forest (**b**), secondary forest (**c**) and pasture (**d**) and monthly mean of the Measurements of Pollution in the Troposphere (MOPITT) CO total atmospheric column data for each analised year. **e** Modelled gross $CO_2$ emissions from forest fires and deforestation. Numbers indicate the total annual emission in Tg $CO_2$ per year for the Brazilian Amazon biome

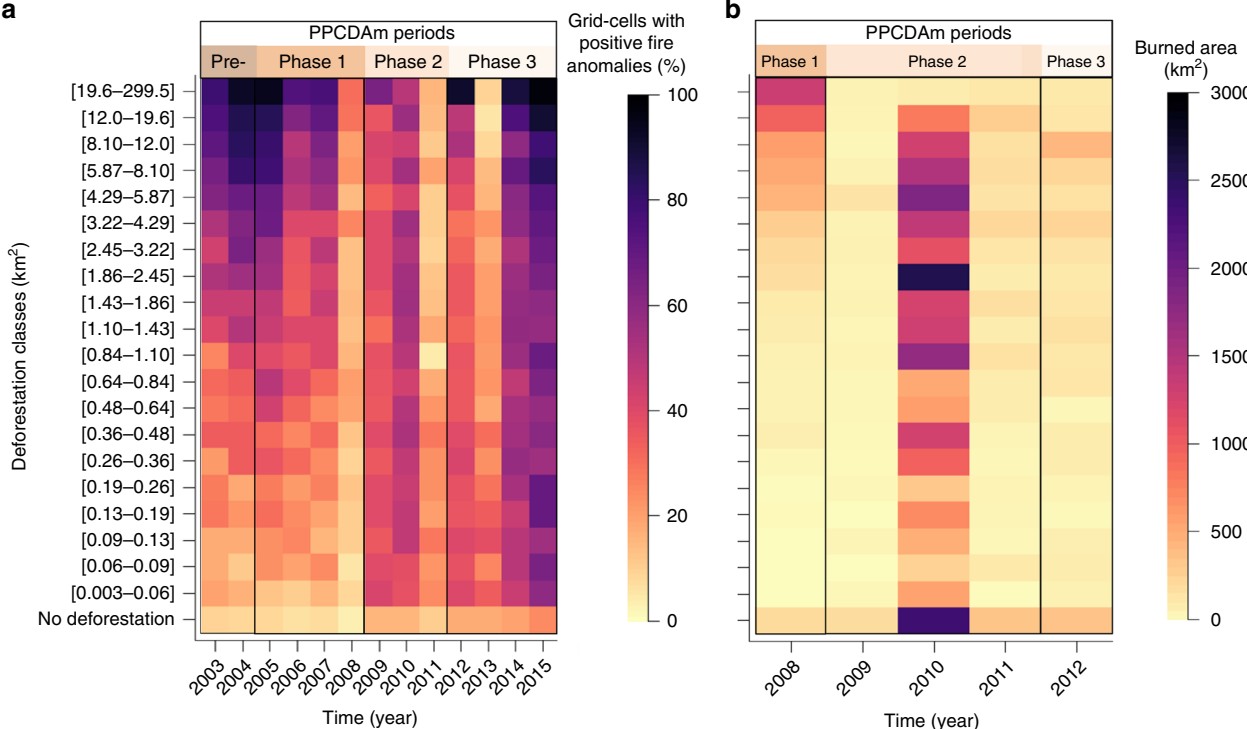

**Fig. 7** Temporal quantification of active fire anomalies and burned area across the deforestation gradient. Temporal patterns of positive active fire anomalies (number of grid cells with positive anomalies) (**a**) and forest plus secondary forest burned area (total area burned) (**b**) across the deforestation gradient. Deforestation categories correspond to the area deforested each year. The figure shows deforestation data classified in 20 percentiles, in addition to one class where deforestation was not observed (no deforestation)

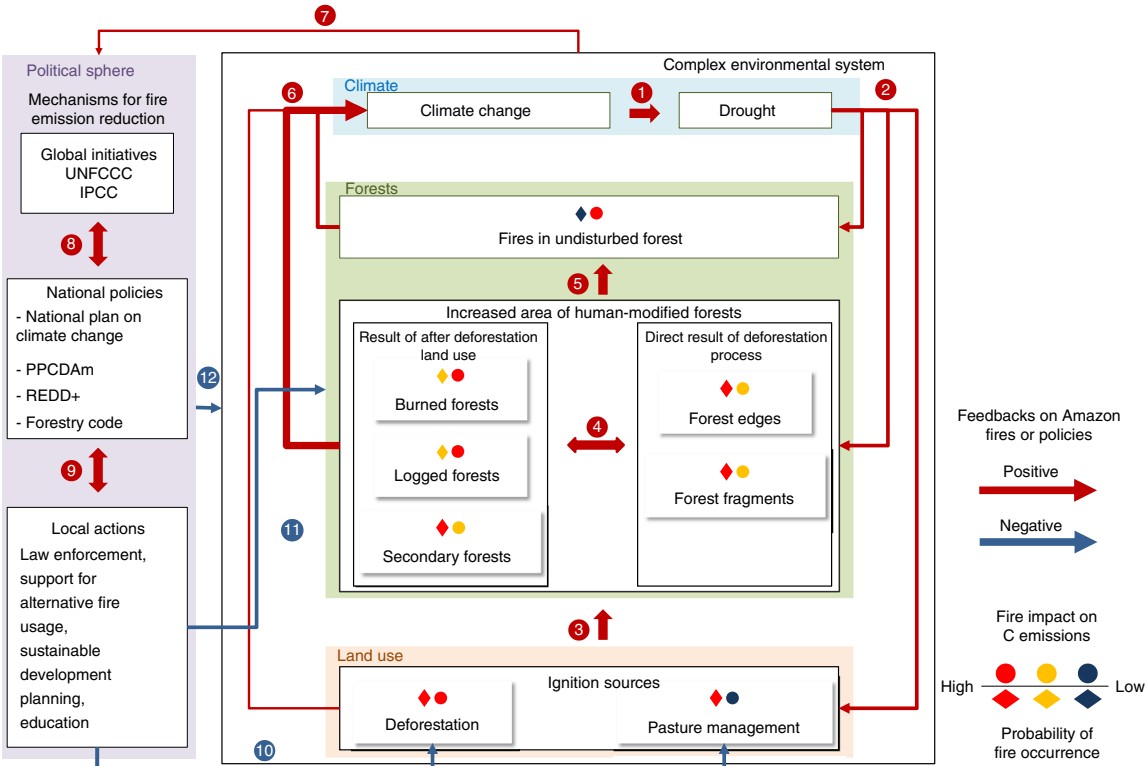

**Fig. 8** Conceptual model of feedbacks between climate, land use, forests and policies and their expected impacts on fire emissions. The system is divided in two large components: the political sphere (purple box) and the complex environmental system. In the political sphere, the mechanisms for fire emission reduction are divided into three levels of organization from global to local. The complex environmental system is divided in three components: climate (blue box), forests (green box) and land use (light brown box). Positive and negative feedbacks among the components are identified by red and blue arrows, respectively. The resulting effects of these feedbacks are described by coloured diamonds for the expected probability of fire occurrence and circles for the potential fire impact on C emissions. Numbers are displayed to assist with the description of the processes depicted in the main text

that areas other than monitored deforested land by PRODES are increasingly burning during extreme droughts.

We suggest that the Brazilian Amazon may be entering a new land use and land cover change phase in which a decoupling between fire-related and deforestation carbon emissions, driven by recurrent 21st century droughts, can undermine the Brazilian achievement of reducing emissions from deforestation. If this new arrangement persists through time, reported national $CO_2$ emission levels[26] to the United Nations Framework Convention on Climate Change (UNFCCC) is expected to be underrated if $CO_2$ emission estimates do not account for non-deforestation fires, especially the forest fires examined in this study.

It is important to note that C emission estimates based exclusively on the deforestation process, not only exclude emissions from forest fires, as reported here, but also ignore emissions from other neglected processes. For instance, repeated clearing and burning of secondary vegetation, which is estimated to have suppressed 25% (38,000 km$^2$) of the total secondary vegetation area from 2008 to 2012[33], is not quantified by the PRODES system. This loss of secondary forest would only affect the net annual emission calculations, with negligible long-term effect on atmospheric $CO_2$ concentrations, because of continuous regeneration of these forests. Additionally, the deforestation of areas smaller than the detection limit of PRODES (6.5 ha)[7], while highly uncertain in magnitude, could have omitted around 9000 km$^2$ of deforested land from 2008 to 2012[34]. The effect of this component on the Brazilian Amazon emission estimates does not surpass the influence of forest fire-associated emissions, as the area of this missing process is only 15% of that from old-growth

and secondary forests burned in the same period. Our analyses, hence, confirm the hypothesis that C emissions from the Brazilian Amazon are increasingly dominated by forest fires during extreme droughts, rather than emissions from fires directly associated with the deforestation process[15].

The processes constraining drought and fires in Amazonia and the potential feedbacks leading to the strengthening or weakening of this association can be summarized as follow (Fig. 8): The predicted augmentation of dry season intensity in Amazonia during the 21st century tends to drive changes in large-scale atmospheric circulation patterns[29], resulting in below-average rainfall over Amazonia (Pathway 1). Consequently, rainfall shortage generates high water deficits. In pathway 2, drought-induced water stress on forests acts negatively on the overall photosynthetic capacity of the system, causing widespread tree mortality[35], leaf shedding and thus increasing fuel availability[16, 36, 37]. As a consequence, forest canopies become more open, boosting incident radiation levels within the canopy and producing rising temperatures[38]. The increased fuel availability exposed to a drier and hotter microclimate pushes natural forests towards a fire-prone system. Droughts can also double the average rate of fire incidence in Amazonia associated with increased persistence of pasture management and deforestation fires[5]. These factors increase the probability of fires to leak from open areas into adjacent human-modified forests (forest edges, fragments, burned, logged and secondary forests), all of which tend to be more susceptible to fire than large blocks of undisturbed primary forests[39, 40] (Pathway 3). Although previous Amazonian-scale quantification (1992–1999) demonstrated that

burned scars from fires were rarely detected in degraded logged forests[41], the observed intensification of 21st century climate extremes and droughts suggest a potential shift toward the previously postulated model[39, 40]. On the basis of this model[39, 40], we suggest that under the new 21st century climate conditions degraded forests may become increasingly dry and susceptible to forest fires. In comparison with intact forests, human-modified forests are characterized by greater canopy opening, larger stocks of dead material, drier microclimate, and lower wood density species[42], which are more susceptible to mortality during droughts[35]. Such characteristics make these forests more flammable and act to increase fire intensity if they do burn, facilitating fire incursion among the different forest types (Pathway 4) and also into undisturbed old-growth forests (Pathway 5). Fires occurring in different land uses and forests have a direct impact on climate change by increasing C and aerosol loads in the atmosphere (Pathway 6). The presence of fire-derived aerosols in the atmosphere can reduce rainfall, by inhibiting surface heating and evaporation, suppressing the formation of convective clouds and by affecting cloud's microphysics by reducing droplet size in comparison to clean air, which in turn inhibits the onset of precipitation[43]. Understanding these processes improves the efficacy of environmental policies for counteracting the observed negative impacts and positive feedbacks driving this complex system (Pathway 7). Information gathered from monitoring the process facilitates global initiatives for developing standardized reporting guidelines and planning financial incentives (Pathway 8), which can positively feedback into national policies, promoting effective local actions (Pathway 9). A well-informed political sphere is able to locally reduce the impacts of land use on fires (Pathway 10) and consequent fire impacts on forests, local economies and human populations (Pathway 11). Finally, structured environmental policies can assist managing the drivers of climate change and land use, alleviating their consequent impacts on tropical countries (Pathway 12).

While extensive advances were made to report emissions from direct land cover conversion[26, 44, 45], such as those from deforestation, efforts still must be made to incorporate into estimates, $CO_2$ losses associated with fires unrelated to the deforestation process, as proposed by the Intergovernmental Panel on Climate Change (IPCC) guidelines for National Greenhouse Gas Inventories[46]. Explicitly accounting for drought-induced forest fire emissions as an additional component of C emissions from deforestation is critical for accurately quantifying the overall Amazonian net C budget.

Even if Brazil achieves the end of Amazonian deforestation[47], pervasive land use activities and the intensification of extreme droughts are likely to increase fire emissions unrelated to the deforestation, risking the stability of forest carbon stocks and undermining the biodiversity co-benefits achievable in carbon conservation schemes, such as reducing emissions from deforestation and forest degradation (REDD+)[48]. The decoupling between PRODES deforestation estimates and other land cover fires, especially forest fires, suggests that $CO_2$ emissions from these processes must be independently quantified for effectively reporting national emissions from LULCC. If the decoupling between drought-induced fires and deforestation emissions is sustained, we urge that policy actions must amplify the focus on new practices of land management prioritizing non-deforestation fire reduction. It is critical to note that if the observed decline trend in deforestation rates is reverted, the additive characteristic of emissions from deforestation-related and non-deforestation fires, could promote a rise in carbon emissions to an unforeseen rate. Governmental intervention to maintain low deforestation levels, manage farming activities and combat fires in very dry years, therefore, is of upmost importance for reducing future C emissions from the Brazilian Amazon.

## Methods

**Sea surface temperature anomalies.** Time series of sea surface temperature anomalies (SSTA) for the Pacific and Atlantic Oceans were obtained from the National Oceanic and Atmospheric Administration (NOAA), Earth System Research Laboratory portal. The Atlantic Multidecadal Oscillation index (AMO)[20], which describes average anomalies of sea surface temperatures (SST) in the North Atlantic basin, typically over 0–70N (https://www.esrl.noaa.gov/psd/data/timeseries/AMO/), is calculated from the Kaplan SST V2 data set as a departure from 1951 to 1980 time period[49]. The Multivariate El Niño Index (MEI)[21] integrates six variables that describe the status of the coupled ocean-atmosphere system over the tropical Pacific from 30N to 30S. The variables integrated into MEI are: (1) sea-level pressure, (2) zonal and (3) meridional components of the surface wind, (4) sea surface temperature, (5) surface air temperature, and (6) total cloudiness fraction of the sky. All seasonal values are standardized with respect to each season and to the 1950–1993 reference period (https://www.esrl.noaa.gov/psd/enso/mei/table.html). The Pacific Decadal Oscillation index (PDO), calculated for the North Pacific Ocean, poleward of 20N, represents the leading principal component of North Pacific monthly sea surface temperature variability for the 1900–93 period[22]. NOAA's National Centers for Environmental Information (NCEI) PDO index, used in this study, is based on NOAA's extended reconstruction of SSTs version 4 (https://www.ncdc.noaa.gov/teleconnections/pdo/).

Global gridded SSTA map for September 2015 (Fig. 1f) was produced using the monthly SSTA NOAA NCEP (National Centers for Environmental Prediction) EMC (Environmental Modeling Center) CMB (Climate Modeling Branch) Global Reyn_SmithOIv2 product (http://iridl.ldeo.columbia.edu/SOURCES/.NOAA/.NCEP/.EMC/.CMB/.GLOBAL/.Reyn_SmithOIv2/). Anomalies are calculated as the departure from the 1971 to 2000 climatology[50]. Details of the geographical locations in which the calculations of oceanic indices are based on (Fig. 1f) follow Enfield et al.[20] for AMO, Wolter and Timlin[51] for MEI and Mantua et al.[22] for PDO.

**Rainfall and maximun cumulative water deficit**. We quantified the intensity and duration of the drought across Amazonia by calculating annual rainfall and maximum cumulative water deficit (MCWD) anomalies (z-scores) for 2005, 2010 and 2015 as the departure from the 2003–2014 mean, normalized by the s.d. ($\sigma$), using TRMM 3B43 7A product from 2003 to 2015 (https://disc.gsfc.nasa.gov/datacollection/TRMM_3B43_7.html)[23]. We have excluded from the long-term mean all the analyzed drought years. We calculated monthly values for both variables based on a average rainfall and MCWD for the whole basin. We also produced pixel-by-pixel mean and standard deviations of monthly precipitation at 0.25° spatial resolution for the Amazon Biome. The cumulative monthly precipitation was estimated in mm month$^{-1}$ considering a 30-day month for all the data sets. The p-values associated with the z-scores (anomalies) calculated in this study were analized statistically considering the standard normal distribution, where positive or negative anomalies ($\sigma$) between: $1.65 \le \sigma < 1.96$ are significant at 90% confidence level, $1.96 \le \sigma < 2.58$ are significant at 95% confidence level and >2.58 are significant at 99% confidence level.

The pixel-based calculation of the annual maximum cumulative water deficit (MCWD) from 1998 to 2015, corresponded to the maximum value of the monthly accumulated water deficit (WD) reached for each pixel within each year[17]. The WD considers the mean evapotranspiration value obtained by ground measurements in different locations and seasons in Amazonia[17]. Hence, based on the approximation that a moist tropical canopy transpires ~100 mm month$^{-1}$, when monthly rainfall ($P$) is less than this value the forest enters into water deficit. The following rule was applied to calculate the WD for each month ($n$) on a pixel-by-pixel basis (each located at $i$ column and $j$ line), with evapotranspiration ($E$), fixed at 100 mm month$^{-1}$:

$$\text{If } WD_{n-1(i,j)} - E_{(i,j)} + P_{n(i,j)} < 0;$$
$$\text{then } WD_{n(i,j)} = WD_{n-1(i,j)} - E_{(i,j)} + P_{n(i,j)};$$
$$\text{else } WD_{n(i,j)} = 0$$

The MCWD was obtained for each pixel as the negative of the minimum value of WD among all the months in each one of the years. The MCWD is a useful indicator of meteorologically induced water stress without taking into account local soil conditions and plant adaptations, which are poorly understood in Amazonia.

**Active fire incidence**. The intensity of fire incidence across the Amazon biome within the limits of the Brazilian Amazon was calculated using active fire pixels or hot pixels data, available from the INPE's Center for Weather Forecasting and Climate studies (CPTEC) fire monitoring system (http://www.inpe.br/queimadas). This data is derived from the MODIS (Moderate Resolution Imaging Spectroradiometer) sensor on board the polar orbiting Aqua satellite, based on its afternoon overpasses[24]. The monthly and annual number of fires was quantified as active fire density (accumulated number of monthly active fire counts) by summing

the daily observations with a nominal 1 km spatial resolution within each grid cell with 0.25° spatial resolution. The thermal anomalies (z-scores) were calculated similarly to the previous data sets.

**Burned area mapping by land cover**. We produced burned area maps from 2008 to 2012, centered in 2010. This period accounted for one year prior and one year after PPCDAm phase 2 (2009–2011). The analysis using burned area information was designed to capture the constant decrease in deforestation rates ($-1773 \pm 507$ km² year⁻¹), including a drought year (2010). This 2nd phase consolidated the PPCDAm programme focusing mainly on environmental monitoring and control.

For mapping burn scars we follow the methodology described by Anderson et al[9]. We used daily surface reflectance products MOD09GA and MOD09GQ as well as 8 day surface reflectance products, MOD09Q1 and MOD09A1, collection 5 from the MODIS data set. The dates of the cloud-free images were selected based on the latest day of the month with nadir view covering the Brazilian Legal Amazon. The burned area maps were based on the linear spectral mixing model (LSMM) applied on three spectral bands[9]: red (band 1, 620–670 nm) and near-infrared (NIR; band 2, 841–876 nm) reflectance bands from MOD09GQ product and the shortwave infrared (SWIR; band 6, 1628–1652 nm) from the MOD09GA product. From the outputs of the LSMM we used the shade fraction image, which contains the relevant information for mapping burnt areas. We first applied a segmentation procedure using a minimum area threshold of 4 pixels (~25 ha). Subsequently, an unsupervised classification was performed followed by a post classification image edition. The post classification image edition was carried out by a skilled human interpreter using the natural colour composites of the corresponding images for comparison, aiming to minimize omission and commission errors normally produced by any automatic classification algorithm.

Subsequently, burn scar maps were combined with the INPE-TerraClass[33] land cover maps (http://www.inpe.br/cra/projetos_pesquisas/dados_terraclass.php) to divide burned scars by land cover classes. For this study we focused on three classes: First, old-growth forest, second, secondary forest and third, pasture. All burn scars occuring in areas mapped as deforested or non-forests in the INPE-TerraClass[33] data set were excluded from the analysis.

Our burnt area estimate was likely to be conservative as the use of MODIS data at 250 m spatial resolution can underestimate the area burned by approximately 25% in relation to manually digitized burn scars based on 30 m spatial resolution Landsat images[52, 53].

**Deforestation rates**. The annual cumulative deforested area from 2003 to 2015 was obtained from INPE's PRODES data set (http://www.obt.inpe.br/prodes/index.php)[7]. These values are Brazilian Government's official estimates of annual deforestation in Amazonia. PRODES uses Landsat-like satellites, with 20–30 m spatial resolution, to quantify the complete conversion of old-growth forests into agricultural uses. PRODES only maps polygons with an area greater than 6.25 ha and does not account for deforestation of secondary regenerating forests. Images from Landsat-5 Thematic Mapper sensor are the most used in the programme. To overcome commonly encountered cloud cover problems, however, other sensors such as the CCD from CBERS-2 and CBERS-2B, LISS-3 onboard Resourcesat-1, and UK-DMC2 images can also be utilized.

**MOPITT data**. The interannual variability of Amazonian biomass burning emissions was analyzed using the record of carbon monoxide (CO) measurements from the MOPITT (Measurements of Pollution in the Troposphere) satellite instrument. The MOPITT instrument incorporates gas-filter correlation radiometers operating in both TIR (thermal-infrared) and NIR (near-infrared) spectral bands[54]. MOPITT began operations in 2000. Results presented in this manuscript are based on the MOPITT Version 6 Level 3 (gridded) monthly mean data set exploiting both TIR and NIR observations[25]. This product has been thoroughly validated using in situ CO measurements acquired from aircraft in diverse locations, including four stations in Amazonia[25, 55]. For each month of the MOPITT mission, a basin-wide CO average was obtained by averaging V6 Level 3 CO total column values over all valid one-degree grid cells within the Amazon Basin boundaries. Monthly mean values were then used to calculate annual means.

**CO₂ emissions from deforestation and forest fires**. Reported data on net CO₂ emissions for the Brazilian Amazon biome from 2003 to 2012 was acquired from the report on annual estimates of greenhouse gas emissions (http://sirene.mcti.gov.br/documents/1686653/1706227/Estimativas+2ed.pdf/0abe2683-e0a8-4563-b2cb-4c5cc536c336) developed by Brazilian's Ministry of Science, Technology and Inovation[26]. The estimates of Brazilian reference levels of CO₂ emissions for the Land Use and Land Cover Change (LULCC) sector follows the IPCC Guidelines for National Greenhouse Gas Inventories[46], the Good Practice Guidance and Uncertainty Management in National Greenhouse Gas Inventories[56] and the Good Practice Guidance for Land Use, Land Use Change and Forestry[57].

To analyze the 13 years contribution of gross CO₂ emissions from forest fires, studied here, in relation to deforestation gross emissions estimates[26], we first performed an ordinary least square regression between MOPITT CO total column and burned forest gross CO₂ emissions. First, we carried out a first-order estimate of burned forest committed gross CO₂ emissions ($F_{emission}$) for the mapped years (2008–2012) based on the following equation 1:

$$F_{emission} = A_{O,S} \times B_{O,S} \times \alpha \times 3.6667 \qquad (1)$$

Where $A_{O,S}$ is the total area burned (km²) for old-growth forests (O) and secondary forests (S) mapped in this study, $B$ is the mean biomass C content of Amazon terra firme forests ($B_O = 16,000$ Mg km⁻² and $B_S = 8000$ Mg km⁻²)[58], $\alpha$ is the emission factor for forests affected by fires ($\alpha = 0.4$)[59] and 3.6667 is the conversion factor from C to CO₂. For this parameterization we did not considered the biomass from selectively logged forests that may have burned. Aboveground carbon in logged forests are estimated to be 35% lower than in undisturbed forests[59]. However, the mean biomass value used in our study[58] includes selectively logged areas from 2000–2004, minimizing the impact of potential biomass overestimation in the calculation of CO₂ emissions. Moreover, the underestimation of burned area by our method, described above, can further couterbalance the overall biomass overestimation effect in the final CO₂ emissions estimates from forest fires.

To estimate gross CO₂ emissions from forest fires for the whole 2003–2015 period, we then performed a least square regression between MOPITT CO total column (independent variable) and $\log_{10}$ transformed burned forest committed gross CO₂ emissions (dependent variable). The resulting equation 2 ($n = 5$, $R^2 = 0.95$, $F = 56.4$, $p = 0.005$), with associated standard error values in parentheses, was used to extrapolate the values for the whole period.

$$\log_{10} F_{emission} = -5.55(\pm 1.02) + 3.91(\pm 0.52) \times MOPPITCO \qquad (2)$$

Subsequently, to estimate gross CO₂ emissions from deforestation for the whole 2003–2015 period, we regressed the reported deforestation gross CO₂ emissions ($D_{emission}$) values from 2003–2012[26] against deforestation rates ($D$)[7], and used the resulting equation 3 ($n = 10$, $R^2 = 0.99$, $F = 2307.1$, $p \le 0.001$) to extrapolate the values.

$$D_{emission} = 93.56(\pm 16.79) + 0.05(\pm 0.001) \times D \qquad (3)$$

We expect that the risk of double counting deforestation emissions as forest fire emissions is negligible, because just minimum fractions of the forest area that have burned are later deforested. Results from a previous analysis[60] demonstrated that only 2.6% of all burned forests between 1999 and 2008 were deforested by 2010.

**Data analysis**. To test our hypothesis on the prevalence of the influence of drought over that from deforestation on fire incidence, consequently impeding a direct reduction of Amazonian C emissions from reducing deforestation rates, we first evaluated the spatial and temporal patterns and trends of rainfall, fires and deforestation using the raw values and anomalies. We then evaluated if the previously attested strong correlation between fire and deforestation[15] was stable during the three phases of the PPCDAm programme. To corroborate this analysis, we removed the influence of annual deforestation rates on total active fire incidence, by calculating the number of active fires divided by the deforested area, and quantified the significance of the relationship between number of active fires per km² deforested and time. This analysis was based on the logic that if deforestation is the main driver of fire incidence, the number of active fires per km² deforested must be kept constant through time, even with the known deforestation decline trend.

Furthermore, we repeated this analysis with the independent MOPITT atmospheric CO data set following the same procedure. We also tested the temporal shifts in the atmospheric CO data normalized by active fires. This analysis followed the logic that if the source of CO₂ emission was not related to deforestation but was instead related to management fires from pastures, which has no net impact on the atmospheric CO₂ concentration, the amount of CO released per fire event should have decreased through time. This is expected as burning in pastures release much less carbon than burning in old-growth forests or forests being converted (deforested).

To disentangle the potential sources of fires (old-growth forests, secondary forests, pastures and deforestation) and validate previous analyses carried out in this study we analyzed burned area data stratified by land cover classes[33] to show changes in the area affected forest fires during droughts and how this information on burned area relates with the atmospheric CO data.

Finally, to demonstrate that areas with high fire activity and forest area burned are not always related to regions with high deforestation activity, we analyzed the temporal patterns of positive active fire anomalies (number of grid cells with positive anomalies) and forest plus secondary forest area burned (total area burned) across the deforestation continuum. Deforestation categories were classified in 20 percentiles corresponding to the area deforested each year, in addition to one class where deforestation was not observed (no deforestation). For

each grid cell, we extracted information on the direction of the active fire anomaly and the burned area. For each deforestation class we then counted the number of grid cell with positive fire anomalies and summed the total area burned.

All maps produced in this study were based on publically available data[7, 23, 50] using ArcGIS 10.

**Data availability**. The data that support the findings of this study are all publicly available from their sources. Processed data, products and code produced in this study are available from the corresponding author upon reasonable request.

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

## Acknowledgements

This work was primarily supported by the Brazilian Research Council (CNPQ—grants 458022/2013-6, 305054/2016-3 and 573797/2008-0), the São Paulo Research Council (FAPESP—grants 2008/57719-9, 2008/58120-3, 2011/51841-0 and 2015/50484-0), the UK Natural Environment Research Council (NERC—NE/F015356/2 and NE/l018123/1) and the Research Council of Norway (grant 230860), with partial support from the NERC consortium grants 'AMAZONICA' (NE/F005806/1) and 'ECOFOR' (NE/K016431/1) and the EU via the 7th grant framework GEOCARBON project (grant number agreement 283080). We thank analysts and scientists at INPE, IPEN, NASA, JAXA and NOAA for providing the freely available datasets. We also thank Luae Andere and Natalia Salazar for technical support on the burn scar maps production and Douglas Gherardi and Thelma Krug for insightful comments on early versions of the manuscript.

## Author contributions

L.E.O.C.A. and L.O.A. designed the research with additional input from M.G.F. L.O.A., M.G.F., F.H.W., C.S,. C.H.L.S.J., L.V., T.M.R., E.A. and M.N.D. prepared the database and processed remote sensing data. L.O.A., M.G.F., M.N.D., F.H.W., C.H.L.S.J. and L.E.O.C.A. analyzed the data, with input from L.G.D. and L.G. L.E.O.C.A., L.O.A., M.G.F., A.P.A., J.B., E.B., L.G.D., L.G., M.G., Y.M., J.A.M., O.P., S.S., M.N.D. and J.B.M. analyzed and interpreted the results. L.E.O.C.A. and L.O.A. wrote the manuscript, with input from all authors.

## Additional information

**Competing interests:** The authors declare no competing financial interests.

