## [Peer Review File · Nature Communications]

Reviewers' comments:

Reviewer #1 (Remarks to the Author):

The manuscript, "21st Century drought-related fires counteract the decline of Amazon deforestation carbon emissions," using a suite of remotely sensed data products and published data offers an argument for revisiting the carbon accounting models currently used in light of expected changes in climate over the Brazilian, Amazon. The primary claim presented is that a new paradigm has emerged where climate induced carbon emissions now exceed LULCC derived emissions. This is an important claim and would have broad interest. The lead author and team are very well published and I have no basic methodological concerns. I am less convinced of the novelty of the primary finding and very much concerned with word choice.

First, the authors' use of LULCC creates confusion in multiple sections of the manuscript. It is entirely reasonable from the standpoint of literature review and community norms to use it, but it creates profound confusion with respect to which changes are driving the observed or reported changes in fires, carbon, or CO. The authors should strongly consider replacing LULCC throughout the manuscript with specific language (i.e. primary forest deforestation, high-grading, secondary forest clearing and regrowth, pasture...). The manuscript needs careful editing for language precision. Most of my initial criticisms of the paper were related to this.

Second, the paradigm shift language may be true, but is not supported by the data and methods presented. The extreme outlier nature of the 2015 drought and the lack of prospective simulation make it difficult to defend. It is a reasonable hypothesis, though. The fact that this manuscript is largely an extension of similar work published previously by the lead author following the 2010 drought makes the methods less novel.

Minor issues:

- 1: The paper did not outline a clear model for how recurrent drought is changing the LULCC paradigm. My sense of the paper is that the carbon outcomes and underlying drivers of carbon emissions are changing.
- 2: I did not understand the claim that fires unrelated to direct land conversion are not explicitly included in carbon inventories. See e.g. doi:10.5194/acp-10-11707-2010 - Global fire emissions and the contribution of deforestation, savanna, forest, agricultural, and peat fires (1997–2009)
- 3: The paper needs to make a better case for why fires per km deforested is a good measure. Over a time-series this seems like a prior-probability issue, particularly since the denominator must decline over time – statistical bounding problem.
- 4: The Pete Richards paper you cite is somewhat controversial and could affect the models presented particularly if uncertainty or error increase over time.
- 5: Lines 183 – 197 should be reviewed and better argued. (the previous item would impact this section as well)

Reviewer #2 (Remarks to the Author):

The manuscript "21st century drought-related fires counteract the decline of Amazon deforestation carbon emissions" by Aragão et al. focuses on the very interesting and relevant topic of what controls carbon emissions in Amazonia. The authors perform a rigorous remote-sensing analysis to quantify drought and fire occurrence across space and time, making a strong case for drought-driven fire regimes in Amazonia. Then the authors make a temporal comparison between fire and deforestation and atmospheric carbon monoxide concentrations to demonstrate fire and associated carbon emissions can be strongly decoupled from deforestation, especially in drought years. Although the drought-fire link has been demonstrated in other studies, the rigor of their analysis combined with the demonstration that drought-driven fire emissions are decoupled from

deforestation make this paper a very strong contribution to the field.

The numbers are also quite striking in their magnitude, with fire-driven C emissions in non-deforested zones amounting to ~50% of the emissions due to land use and land cover change demonstrating the necessity to quantify fire rigorously when evaluating carbon sinks in forests and regional inventories of the carbon balance in secondary vegetation.

The component of the analysis that I found most convincing was the spatially explicit comparison between drought activity and fires. The subsequent arguments that fires were largely decoupled from deforestation were logical, but not as quantitatively rigorous as I believe they could be. Specifically, the shift from a spatially-explicit comparison to a temporal comparison between trends in deforestation vs. fire, is not as strong as a spatially-explicit analysis demonstrating that areas with high fire activity are not always related to high deforestation activity. If this analysis is possible, it would improve the robustness of their conclusions.

For example, the authors could perform a spatially explicit analysis of the deforestation-fire relationship from the past, and then apply that correlation to what would be expected in 2015 given the deforestation scenarios that would then be contrasted with the actual pattern of burned area. This would be a convincing argument for disconnect between fire activity in deforestation. However, I am unfamiliar with the deforestation datasets and their degree of accuracy, so am not entirely sure of this feasibility.

All in all this study is strong and novel and should be acceptable for publication with minor revisions. See below for line-by-line comments.

Lines 75-85 are great, very clearly lays out the importance of fires and potential shift to the dominant form of carbon emissions.

Comparing Fig 2c vs. f, seems that fire activity in the eastern amazon is definitely different from past droughts, but also a broad area across the basin is burning going south west, what might these be due to?

Lines 122-128. This paragraph is great, and Figure E4 is very compelling, but I wonder if there could potentially be anything gained by presenting a correlation coefficient in the text to summarize the mean (or median) coefficients so that the reader can be convinced more easily.

Line 137, a $p < 0.1$ reference may come under criticism, perhaps provide another component or descriptive statistic to this (eg., providing the F statistic if you performed an ANOVA); also, perhaps add in the p-values separately for the tests between 2015 vs. 2005 and 2010, if these were explicitly performed.

Line 145, By reading only the main text, I am confused about how the statistical analysis of anomalies was performed. Since the analysis of anomalies is extensive in the main text, I think there should be a sentence describing how the anomalies were statistically defined. E.g., Line 139 has >3 -sigma and Line 146 is >2 -sigma.

Line 149, "distant from the main deforestation epicenters, perhaps add in some description or reference to the exact distance for those that do not know exactly where the deforestation "epicenters" are. E.g., adding an additional map to Fig 2, or a reference to a supplemental figure.

Lines 203-205: is this t-test using year as replicate for the comparison?

Reviewer #3 (Remarks to the Author):

This is a very interesting and important analysis. The authors of this paper make two important assessments related to Amazon fires and greenhouse gas emissions. First, an analysis of meteorological and climatological factors explain an increase in the number of drought-related fires in the Amazon basin forests. Second, while deforestation rates have been declining in recent years, drought-related fires are offsetting any decline in GHG emissions, resulting in more emissions than would otherwise be estimated from deforestation alone. As a matter of relevance, the authors point out how this situation would affect estimates being reported to international conventions on climate change. To demonstrate the first claim the authors do an excellent job constructing the mechanism for rainfall and maximum cumulative water deficits based on

observations and models. This part of the analysis seems clear. The second part of the analysis is less coherent and conclusive.

Observations of fires in the Amazon have been made for many years. The analysis rightly uses an excellent long term dataset of fires from the CPTEC fire monitoring system. The method for detection is solid and the fire counts as expressed in the paper are firm. The more difficult issue is ascertaining whether the observed fires occur in areas of high biomass (primary and secondary forests) or on areas of low biomass (pastures). More important than biomass differences (which influence the magnitude of the GHG emissions) is the fact that fires on pastures are usually associated with pasture management, and in the course of a normal land use grass burned in one year regrows in following years, with the net flux of CO₂ being close to zero, thus not an important measurement.

The claim in this paper is that a large number of observed fires in drought years occur in forests or secondary forests. It is laudable that the authors recognize that they need to make a distinction between fires on pastures and fires in forests. But there are several important unsubstantiated claims and some key uncertainties. First, the mapping of fires is based on the CPTEC 1 km fire hot spots. In terms of measuring overall drought-related increases in fire occurrence, this is fine. However, at 1 km aggregation, it is extremely difficult to know with certainty that a fire observation is located in high biomass (ie not in a pasture). The analysis method does indeed review a 5-year window of time that spans both the period of the second PPCDAm program (which controlled deforestation) and a drought year, where the study mapped burn scars onto land cover maps. However, this reviewer has some concerns. First, the land cover maps that are used are presumably from the PRODES program, and these do not generally map regenerating forests nor degraded forests (ie forests thinned by selective logging), so it may be difficult to pin point the exact location of a fire hot spot and document that it did not occur in pasture; perhaps this was done but the paper is unclear on details (NB: the referenced Extended data Fig 8 was not included in the manuscript).

More important, the analysis must clearly distinguish a fire scar that was not also associated with deforestation; more often than not forest land that is converted to non forest land by deforestation is also associated with fire – and so deforestation can produce a burn scar and are thus not two separate emission events. The method to discriminate a fire scar associated with deforestation from a fire scar that is not associated with deforestation is detail work that is not adequately described in the manuscript. At any rate the fire scar analysis does not cover the drought year of 2015 which is an important gap in their model.

Lastly the authors would do well to know better the deforestation and land cover change literature for the Amazon. The analysis does not seem to include, or reflect, the importance of areas of forest that may be degraded by selective logging. Although they recognize that fires could occur in secondary forests, they omit a discussion of degraded forest, which actually may be more important. This is important because there has been some recent debate over the coupling of selective logging and fire. The first studies, such as Cochrane et al (1999, 2001) argued that selective logging changes local moisture conditions, making logged forest susceptible to drying and then fire. In this model, the fire source would be a proximate deforestation event or pasture fire which would “escape” to completely or significantly burn dry, degraded logged forest. This model, however, was based on studies over limited geographic scale, generally a few satellite images in a region around Paragominas. When the entire Amazon was analyzed by Matricardi et al. (2013) the coincidence of forest degradation by logging and subsequent fire did not exist. Most fires in forests did not occur on land that was degraded by logging. Most fires were found in forest edges and intact forests. The interesting and important question here is whether increased drought and drought intensity could be catalyzing a change in this decoupled trend observed by Matricardi in contrast to the coupled condition postulated by Cochrane. There could be a trend now being observed by these authors toward much more fire in these degraded forests that are now increasingly dry and susceptible. Hence, while Matricardi observed only few cases where fire occurred following degradation by logging, climate and drought are changing conditions

significantly. The authors need to consider this potential because it would suggest a shift in the recent patterns observed by Matricardi et al (2013) toward the model that had been locally idiosyncratic proposed by Cochrane et al (1999, 2001), and provide a precise mechanism to explain the dynamics of a situation where there is less outright deforestation yet increasing forest fire. This may be difficult since the PRODES program does not map forest degradation, but also be very interesting because it would suggest the increase in fire is indeed related to land use change – but greatly enhanced by climate change.

This study is completely reasonable in its assumptions and conclusions, and indeed I too would be prone to believe that climate change related drought would increase fire emissions even while outright deforestation rates are declining. The MOPITT CO data are important in this regard. I just want the authors to nail down the specifics and details in a more convincing and mechanistic way (e.g. explain how standing forests can burn and support their statement that fires “leak” to surrounding forests).

Some attention also needs to be given to the importance, or not, of increasing fire sizes vs increasing number of fires.

Lastly there are many small technical and editing issues – the word data is plural, Extended data Fig 8 is missing, typos (e.g. MODIS product names).

Mark A. Cochrane, Ane Alencar, Mark D. Schulze, Carlos M. Souza Jr., Daniel C. Nepstad, Paul Lefebvre, Eric A. Davidson. 1999. Positive feedbacks in the fire dynamic of closed canopy tropical forests, *Science* (284) 5421:1832-1835, 11 JUN 1999

Mark A. Cochrane. 2001. Synergistic interactions between habitat fragmentation and fire in evergreen tropical forests, *Conservation Biology* 15: 1515–21.

Eraldo A.T. Matricardi , David L. Skole , Marcos A. Pedlowski and Walter Chomentowski (2013) Assessment of forest disturbances by selective logging and forest fires in the Brazilian Amazon using Landsat data, *International Journal of Remote Sensing*, 34:4, 1057-1086, DOI: 10.1080/01431161.2012.717182

Responses to reviewers of the manuscript entitle:
**“21st Century drought-related fires counteract
the decline of Amazon deforestation carbon
emissions”**

By Aragão L.E.O.C. and co-authors

First of all, we thank all three Reviewers for their comprehensive review and solid comments. We really appreciated the suggestions and comments. We have addressed them in full, and feel that this new version of the manuscript is clearer for Nature Communications readers. We also believe the results and conclusions of the paper are now more robust.

Below, we repeat all Reviewers' comments and reply to the concerns one by one. Each comment is numbered, using a continuous sequence and with our responses in bold.

Reviewer #1

1. The manuscript, “21st Century drought-related fires counteract the decline of Amazon deforestation carbon emissions,” using a suite of remotely sensed data products and published data offers an argument for revisiting the carbon accounting models currently used in light of expected changes in climate over the Brazilian Amazon. The primary claim presented is that a new paradigm has emerged where climate induced carbon emissions now exceed LULCC derived emissions. This is an important claim and would have broad interest. The lead author and team are very well published and I have no basic methodological concerns.

Answer 1: We are glad that the reviewer liked the narrative of the paper, its message and the methodology used. We thank the reviewer for the comprehensive revision of the manuscript. We have incorporated all suggestions into the revised paper. We are sure that the revisions based on the reviewer comments have strengthened the new version of the paper.

2. I am less convinced of the novelty of the primary finding

Answer 2: The reviewer questions the novelty of the primary finding referring to “The primary claim presented is that a new paradigm has emerged where climate induced carbon emissions now exceed LULCC derived emission”.

As far as we know, this is the first study to show that drought-induced forest fire emissions can exceed deforestation emissions under extreme drought conditions. We believe the reviewer questions the novelty of the “new paradigm”, based on previous publications from our group. To clarify, previously we showed increased active fire trends in the majority of grid-cells with decreased deforestation trends,

hypothesising that these patterns *could* lead to increased fire emissions even if deforestation declined (Aragão et al. 2010a, *Science*). The hypothesis was probed with a modelling exercise that indicated low probability of this pattern to be a consequence of pasture fires (Aragão et al. 2010b, *Science*), that have no long-term impact on the atmospheric C balance, as pastures tend to recover their lost biomass from fires very quickly after the event. These previous analyses were carried out using exclusively active fire data (hot pixels), which are limited in providing an accurate information on the fire source. In the current analysis, on the other hand, we used a suite of independent datasets to carry out the research, including specially data on burned area that allowed the quantification of emissions.

Therefore none of these previous studies explicitly tackled the problem with actual observations on active fires, atmospheric CO concentration data and observations of forest burned area. So, while this study builds on previous work, the main results and conclusions are completely novel. It provides the first first-order estimates of forest fire-associated CO₂ emissions, and shows that these emissions can surpass emissions from deforestation during drought years. This paper therefore validates previous hypothetical conjecture.

Based on the reviewer comment, however, we have decided to reduce the focus of the text on the “new paradigm” and bring it as a plausible hypothesis (comment 5, Reviewer 1) in the discussion of the results. Our decision is based on the fact that despite being confident that our observations point towards the proportional dominance of forest fire emissions in relation to deforestation emissions, we cannot of course guarantee that this shift will be sustained in the future (which would be required for it to be considered a new paradigm).

References:

Aragão, L. E. O. C.; Shimabukuro, Y. E. The Incidence of Fire in Amazonian Forests with Implications for REDD. *Science*, 2010a, 328 (5983), 1275-1278.

Aragão, L. E. O. C.; Shimabukuro, Y. E. Response to Comment on “The Incidence of Fire in Amazonian Forests with Implications for REDD”. *Science*, 2010b, 330 (6011), 1627.

3. ...and very much concerned with word choice. First, the authors’ use of LULCC creates confusion in multiple sections of the manuscript. It is entirely reasonable from the standpoint of literature review and community norms to use it, but it creates profound confusion with respect to which changes are driving the observed or reported changes in fires, carbon, or CO. The authors should strongly consider replacing LULCC throughout the manuscript with specific language (i.e. primary forest deforestation, high-grading, secondary forest clearing and regrowth, pasture...).

Answer 3: We agree with the reviewer’s observation. We followed the recommendation of the reviewer and have now attempted to be more specific in defining each process. We have changed the terms throughout the text.

4. The manuscript needs careful editing for language precision. Most of my initial criticisms of the paper were related to this.

Answer 4: The manuscript was revised by our English speaking' co-authors.

5. Second, the paradigm shift language may be true, but is not supported by the data and methods presented. The extreme outlier nature of the 2015 drought and the lack of prospective simulation make it difficult to defend. It is a reasonable hypothesis, though.

Answer 5: We agree with the reviewer. For this reason, we now present it as a hypothesis, as suggested by the reviewer, in the discussion and conclusions section. Please refer to the last paragraph of our answer 2.

6. The fact that this manuscript is largely an extension of similar work published previously by the lead author following the 2010 drought makes the methods less novel.

Answer 6: Please see the details of our response to this comment in the third paragraph of answer 2.

Minor issues:

7: The paper did not outline a clear model for how recurrent drought is changing the LULCC paradigm. My sense of the paper is that the carbon outcomes and underlying drivers of carbon emissions are changing.

Answer 7: Yes. We agree with the reviewer. This was an excellent observation and we have now added a new figure (Figure 8) with a conceptual model of the full process as suggested with a paragraph in the discussion depicting the whole process investigated in the paper.

8: I did not understand the claim that fires unrelated to direct land conversion are not explicitly included in carbon inventories. See e.g. doi:10.5194/acp-10-11707-2010 - Global fire emissions and the contribution of deforestation, savanna, forest, agricultural, and peat fires (1997–2009)

Answer 8: We understand the confusion. The text was not fully accurate and we have changed the statement. We are aware of the paper suggested by the reviewer and have already cited this paper in the previous version of the manuscript. We have changed the sentence for clarity. Now the sentence reads:

“Yet, emissions from drought-induced forest fires unrelated to direct land cover conversion are usually not explicitly included in national-level carbon emissions inventories.”

The main point of this sentence is to highlight to readers that emissions unrelated to deforestation are not normally included in official (Governmental) national level inventories, even though global estimates are available in the academic literature.

9: The paper needs to make a better case for why fires per km deforested is a good measure. Over a time-series this seems like a prior-probability issue, particularly since the denominator must decline over time – statistical bounding problem.

Answer 9: We understand the concern of the reviewer and add more explanation justifying the use of fires per km deforested and other measures in a new Data analysis section in the Methods.

We avoided using this as a unique measure; instead we used a suite of evidence, including the measure cited by the reviewer, to test our hypothesis. We present the methods in a coherent sequence starting with the evaluation of patterns and trends using the raw fire values and anomalies. Subsequently, to remove the effect of deforestation on fire patterns, we divided the number of fires per km² deforested. It is important to evaluate this metric because if fires are directly responding to the deforestation process, as previously attested by the strong relationship between fire and deforestation (Aragão et al. 2008), the number of fires normalized by the area deforested must be kept constant through time, even with the known deforestation decline trend. This was not the case in our study.

Then, we repeated this analysis with an independent dataset focussing on atmospheric CO per km deforested. Similarly, if CO concentration was directly following emissions from deforestation, the ratio CO/deforestation must be kept constant through time.

Moreover, we also tested the temporal shifts in atmospheric CO data normalized by active fires. This analysis followed the logic that if the source of CO₂ emission was not related to deforestation, but was instead related to management fires from pastures, which has no net impact on the atmospheric CO₂ concentration, then the amount of CO released per fire count should have decreased through time. This is because burning in pastures releases much less carbon than burning in old-growth forests or forests being converted (deforested).

Finally, to make a robust affirmation about the process we used burned area data. This, first, indicates the magnification of forest fires during droughts, and second demonstrates a firm link between forest burned area and atmospheric CO data.

In sum, we believe that the measure cited by the reviewer is one of a suite of measurements that together support our conclusions. Removing this measure would not affect the conclusion, but maintaining this analysis reinforces the evidence-base for our conclusions.

10: The Pete Richards paper you cite is somewhat controversial and could affect the models presented particularly if uncertainty or error increases over time.

Answer 10: This is an important comment. We agree that Richards et al. paper is controversial, mainly because of the lack of in-depth explanation for the reader about the purpose, conception and precise technical characteristics of each deforestation system, allowing the readers to have a consistent interpretation of the data. However, their data analysis is not invalid and as far as we know it is the only recent paper that quantifies the differences between PRODES and GFC dataset for the Amazon biome using Hansen's Global Forest Cover (GFC) data. Our use of this paper is restricted to a single result, and we only use this one number provided by Richard et al. in figure S7, which quantifies the potential differences between PRODES and GFC dataset. The value of the difference, removing the systematic bias between the two datasets is 9,000 km² as stated in our text.

We know that the higher forest loss quantified in the GFC dataset corresponds to processes not intended to be mapped by PRODES, such as: small deforestation (<6.5ha), deforestation of secondary forests and forest plantations, degradation processes and deforestation occurring in areas considered as non-forest by PRODES. Because of the amount of possible sources of C emissions in this value, we know that this is an overestimate of the emissions from small deforestation + secondary forest deforestation, which are the two unknown but relevant processes for our analysis.

Yet even with the possibility of uncertainty increasing through time, which was an important point made by the reviewer, the additional C sources estimate is likely to be minor: the total difference of 9,000 km² forest loss from 2008-2012 corresponds to only 15% of the total area burned in old-growth forests and secondary forests during the same period (as quantified in our study). Therefore, the deforestation of secondary forests and deforested areas <6.5ha are not expected to affect our conclusions as: (1) these processes have a small contribution to the gross C source in comparison to the quantified burned area and (2) do not overcome the direct gross emissions from forest fires.

We believe, however, that it is important to clearly show in the text these missing carbon sources, providing the reader with a number for reference. For this reason we cite Richards et al.

11: Lines 183 – 197 should be reviewed and better argued. (the previous item would impact this section as well)

Answer 11: We have improved the description and rationale behind these analyses in the methods section. We believe that with this new section on Data Analysis the text is clearer.

Reviewer #2 (Remarks to the Author):

12. The manuscript “21st century drought-related fires counteract the decline of Amazon deforestation carbon emissions” by Aragão et al. focuses on the very interesting and relevant topic of what controls carbon emissions in Amazonia. The authors perform a rigorous remote-sensing analysis to quantify drought and fire occurrence across space and time, making a strong case for drought-driven fire regimes in Amazonia. Then the authors make a temporal comparison between fire and deforestation and atmospheric carbon monoxide concentrations to demonstrate fire and associated carbon emissions can be strongly decoupled from deforestation, especially in drought years. Although the drought-fire link has been demonstrated in other studies, the rigor of their analysis combined with the demonstration that drought-driven fire emissions are decoupled from deforestation make this paper a very strong contribution to the field.

Answer 12: We are glad that the reviewer liked the main message in the paper and the methods used to reach a consistent conclusion. We thank the reviewer for the comprehensive revision of the manuscript. We have addressed all suggestions in the revised paper. We are sure that the revisions based on the reviewer comments have strengthen the new version of the paper.

13. The numbers are also quite striking in their magnitude, with fire-driven C emissions in non-deforested zones amounting to ~50% of the emissions due to land use and land cover change demonstrating the necessity to quantify fire rigorously when evaluating carbon sinks in forests and regional inventories of the carbon balance in secondary vegetation.

Answer 13: Yes. This is a good point and indeed important for increasing the accuracy of the Amazonian net C budget. We have added a sentence in the conclusions that states: “Explicitly accounting for drought-induced forest fire emissions as an additional component of C emissions from deforestation is critical for accurately quantifying the overall Amazonian net C budget.”

14. The component of the analysis that I found most convincing was the spatially explicit comparison between drought activity and fires. The subsequent arguments that fires were largely decoupled from deforestation were logical, but not as quantitatively rigorous as I believe they could be. Specifically, the shift from a spatially-explicit comparison to a temporal comparison between trends in deforestation vs. fire, is not as strong as a spatially-explicit analysis demonstrating that areas with high fire activity are not always related to high deforestation activity. If this analysis is possible, it would improve the robustness of their conclusions.

Answer 14: Yes. This was an excellent suggestion by the reviewer. We were able to demonstrate that areas with high fire activity are not always related to high deforestation activity by performing a new spatial-temporal analysis (Figure 7) using our observed data. Our new figure shows the temporal patterns of positive active fire anomalies (percent of grid-cells with positive anomalies) and forest plus secondary forest burned area (total area burned) across the deforestation gradient. Deforestation categories correspond to the area deforested each year. The figure shows deforestation data classified in 20 percentiles (km²), in addition to one class where deforestation was not observed (no deforestation).

It is clear by analysing this figure that: (1) areas with high deforestation rates normally maintain high fire levels, as expected, but (2) we also observe that under low deforestation rates the number of grid-cells with positive fire anomalies and burned area increases from 2003, the beginning of the period analysed to 2015, the end year. (3) Forest burned area clearly increases massively during drought, while also (4) during droughts there is even an increase of burn area in grid-cells with no deforestation!

We believe that this analysis substantially improves the robustness of our conclusions.

15. For example, the authors could perform a spatially explicit analysis of the deforestation-fire relationship from the past, and then apply that correlation to what would be expected in 2015 given the deforestation scenarios that would then be contrasted with the actual pattern of burned area. This would be a convincing argument for disconnect between fire activity in deforestation. However, I am unfamiliar with the deforestation datasets and their degree of accuracy, so am not entirely sure of this feasibility.

Answer 15: Please see previous answer. We did not try to perform the suggested analysis in comment 15, because, despite being an interesting way to tackle the problem, we think our deforestation and fire time series are still too short to provide a robust estimate based on the regression. This would imply in a fragile model calibration.. Finally, as fire and deforestation are becoming less correlated, we would likely have several grid-cells with model parameters that were not statistically significant.

In summary, while we appreciated the ideas here, we decided to follow the previous suggestion of the reviewer and use the actual observed data for the analysis.

16. All in all this study is strong and novel and should be acceptable for publication with minor revisions. See below for line-by-line comments.

Answer 16: We have implemented all suggestions and hope the paper is now in an acceptable standard for publication in Nature Communications.

17. Lines 75-85 are great, very clearly lays out the importance of fires and potential shift to the dominant form of carbon emissions.

Answer 17: Thank you.

18. Comparing Fig 2c vs. f, seems that fire activity in the eastern amazon is definitely different from past droughts, but also a broad area across the basin is burning going south west, what might these be due to?

Answer 18: These are also areas exposed to drought with the presence of ignition sources. Despite not being located in the epicentres of the drought, which cover central and eastern Amazonia, these areas are also exposed to MCWD larger than 1σ in relation to the long-term mean.

19. Lines 122-128. This paragraph is great, and Figure E4 is very compelling, but I wonder if there could potentially be anything gained by presenting a correlation coefficient in the text to summarize the mean (or median) coefficients so that the reader can be convinced more easily.

Answer 19: We have merged figure 1g with extended figure 4 by adding the gridded data showing the pixel-based correlation coefficients across the entire area analysed. We are now using this figure as Figure 2 in the body of the text. We believe it makes more sense to display the correlation values in the figure, as the reader can check where the significant values are. This provides a better idea on how variable droughts are in Amazonia and where are the main differences among the droughts.

20. Line 137, a $p < 0.1$ reference may come under criticism, perhaps provide another component or descriptive statistic to this (eg., providing the F statistic if you performed an ANOVA); also, perhaps add in the p-values separately for the tests between 2015 vs. 2005 and 2010, if these were explicitly performed.

Answer 20: This p –value refers to the significance of the MCWD anomaly calculated for each grid-cell. So, for each grid-cell we calculated z-scores, which

are the anomalies in units of standard deviation. This value corresponds to the departure of a specific value in time from the long-term mean, normalized by the standard deviation of the long-term mean. Based on the normal distribution curve, a p-value of 0.1, corresponds to a standard deviation (σ) ≥ 1.65 .

We are not comparing the values between 2015, 2010 and 2005. We are counting the number of cells that have the anomalies (z-scores) $\sigma \geq 1.65$.

We have inserted this information in the text and methods section, where we have a modified paragraph:

“We quantified the intensity and duration of the drought across Amazonia by calculating annual rainfall and MCWD anomalies (z-scores) for 2005, 2010 and 2015 as the departure from the 2003–2014 mean, normalized by the standard deviation (σ), using TRMM 3B43 7A product from 2003 until 2015. We have excluded from the long-term mean all the analysed drought years. We calculated monthly values for both variables based on average rainfall and MCWD for the whole basin. We also produced pixel-by-pixel mean and standard deviation of monthly precipitation at 0.25° spatial resolution for the Amazon Biome. The cumulative monthly precipitation was estimated in mm month⁻¹ considering a 30-day month for all the datasets. The p-values associated to the z-scores (anomalies) calculated in this study are associated with the standard normal distribution, where anomalies (σ) between: (1) $1.65 \leq \sigma < 1.96$ are significant at 90% confidence level, (2) $1.96 \leq \sigma < 2.58$ are significant at 95% confidence level and (3) > 2.58 are significant at 99% confidence level.”

21. Line 145, By reading only the main text, I am confused about how the statistical analysis of anomalies was performed. Since the analysis of anomalies is extensive in the main text, I think there should be a sentence describing how the anomalies were statistically defined. E.g., Line 139 has >3-sigma and Line 146 is >2-sigma.

Answer 21: This is explained in the methods section. The format of this journal requires placing the methods section in the end of the main text. We have now added a reference to the methods the first time the anomaly values appear in the text.

22. Line 149, “distant from the main deforestation epicenters, perhaps add in some description or reference to the exact distance for those that do not know exactly where the deforestation “epicenters” are. E.g., adding an additional map to Fig 2, or a reference to a supplemental figure.

Answer 22: We have added the approximated location of the arc of deforestation in Supplementary Figure 2.

23. Lines 203-205: is this t-test using year as replicate for the comparison?

Answer 23: Yes.

Reviewer #3 (Remarks to the Author):

24. This is a very interesting and important analysis. The authors of this paper make two important assessments related to Amazon fires and greenhouse gas emissions. First, an

analysis of meteorological and climatological factors explain an increase in the number of drought-related fires in the Amazon basin forests. Second, while deforestation rates have been declining in recent years, drought-related fires are offsetting any decline in GHG emissions, resulting in more emissions than would otherwise be estimated from deforestation alone. As a matter of relevance, the authors point out how this situation would affect estimates being reported to international conventions on climate change.

Answer 24: We thank the reviewer for the positive feedback and the comprehensive revision of the manuscript. We have incorporated all suggestions into the revised paper. We are sure that the revisions based on the reviewer comments have strengthen the new version of the paper.

25. To demonstrate the first claim the authors do an excellent job constructing the mechanism for rainfall and maximum cumulative water deficits based on observations and models. This part of the analysis seems clear.

Answer 24: Thank you.

26. The second part of the analysis is less coherent and conclusive. Observations of fires in the Amazon have been made for many years. The analysis rightly uses an excellent long term dataset of fires from the CPTEC fire monitoring system. The method for detection is solid and the fire counts as expressed in the paper are firm.

Answer 26: Thank you for highlighting the robustness of the dataset used. This is truly important for achieving high quality results and strong conclusions. We will reply to all concerns of the reviewer in relation to the second part of the paper point by point below.

27. The more difficult issue is ascertaining whether the observed fires occur in areas of high biomass (primary and secondary forests) or on areas of low biomass (pastures). More important than biomass differences (which influence the magnitude of the GHG emissions) is the fact that fires on pastures are usually associated with pasture management, and in the course of a normal land use grass burned in one year regrows in following years, with the net flux of CO₂ being close to zero, thus not an important measurement.

Answer 27: We agree with the reviewer. Because of these difficulties we used a suite of measurements and analyses to produce multiple lines of evidence to provide a clear answer to the question: What are the main sources of C emissions within the land cover gradient?

We first show that the relationship between fire and deforestation is weakening. So, there are fires occurring in other land cover types that are not associated to deforestation. Then, in the text we state:

“The observed disassociation between deforestation and active fire incidence can be related to increased fire incidence in either (i) already deforested land covered by pastures (no net impact on the atmospheric C burden) or (ii) forested areas dominated by woody vegetation.”

So, we test this by first checking the CO concentration. CO in the atmosphere is a proxy for biomass burning. Therefore, if CO concentration per number of active

fires were decreasing through time, it would indicate that active fires would be releasing less carbon. So, the only plausible interpretation would be that pastures would be the main source of fires, as fires occurring in pastures release much less carbon than forest fires. Subsequently, we evaluated burned area data stratified by land cover to check the contribution of pasture area burned in relation to forest area burned. In fact, most of the burn scars in the 2010 drought year occurred in forests, moreover the differences between pasture fires and forest fires in non-drought years are small. Finally, MOPITT CO data, a proxy for fire C emissions, correlates strongly with forest fires.

Taken together, these provide strong evidence pointing towards a dominance of fires in high-biomass land covers and not in pastures.

28. The claim in this paper is that a large number of observed fires in drought years occur in forests or secondary forests. It is laudable that the authors recognize that they need to make a distinction between fires on pastures and fires in forests. But there are several important unsubstantiated claims and some key uncertainties. First, the mapping of fires is based on the CPTEC 1 km fire hot spots. In terms of measuring overall drought-related increases in fire occurrence, this is fine. However, at 1 km aggregation, it is extremely difficult to know with certainty that a fire observation is located in high biomass (ie not in a pasture).

Answer 28: We agree and that's why we used several datasets. Please, refer to the previous answer for a detailed explanation. Please also see Reviewer 1 - Answer 9.

29. The analysis method does indeed review a 5-year window of time that spans both the period of the second PPCDAm program (which controlled deforestation) and a drought year, where the study mapped burn scars onto land cover maps. However, this reviewer has some concerns. First, the land cover maps that are used are presumably from the PRODES program, and these do not generally map regenerating forests nor degraded forests (ie forests thinned by selective logging), so it may be difficult to pin point the exact location of a fire hot spot and document that it did not occur in pasture; perhaps this was done but the paper is unclear on details (NB: the referenced Extended data Fig 8 was not included in the manuscript).

Answer 29: The reviewer is correct about the PRODES data, but we have not used this dataset for decomposing the burn scars into land cover classes. In the methods we have a section "Burned area mapping by land cover" where we state:

"Finally, the burn scar maps were combined with the INPE-TerraClass land cover map to decompose burn scars by land cover classes. For this study we focused on three classes: (1) forest, (2) secondary forest and (3) pasture".

It is important to highlight that the burn scar maps were not generated based on fire hot spot data. We have produced these spatial-explicit maps based on reflectance data from MODIS as detailed in the Methods. Therefore, we have the accurate location of burned polygons, which allows us to intersect this data with the land cover map.

INPE-TerraClass is a new freely available product that provides information on the land cover within the limits of the PRODES deforestation mask. So, it actually

looks inside each deforested polygon to map the different land cover classes. TerraClass data are available at:

http://www.inpe.br/cra/projetos_pesquisas/dados_terraclass.php

In summary, we produced the burn scars maps based on MODIS reflectance data as detailed in methods section “Burned area mapping by land cover” and then used the TerraClass map stratify the burned classes. We improved the details in the methods section.

Extended Figure 8 was indeed missing but this was a mistake in the previous version of the paper.

30. More important, the analysis must clearly distinguish a fire scar that was not also associated with deforestation; more often than not forest land that is converted to non forest land by deforestation is also associated with fire – and so deforestation can produce a burn scar and are thus not two separate emission events. The method to discriminate a fire scar associated with deforestation from a fire scar that is not associated with deforestation is detail work that is not adequately described in the manuscript. At any rate the fire scar analysis does not cover the drought year of 2015 which is an important gap in their model.

Answer 30: The reviewer is correct. However, by using the INPE-TerraClass maps and PRODES annual deforestation maps to intersect the burn scar maps we were able to remove all fires that occurred in areas detected as deforestation.

It is important to note that we know that “Only 1 per cent of understorey burned area in 1999–2007 was deforested within 3 years, and 3.8 per cent of forests burned between 1999 and 2005 were deforested within 5 years (Morton et al. 2013)” So, only a small fraction of the forests burned are deforested afterwards.

The description of the method for mapping burned scars and subset by land cover classes is describe in the section “Burned area mapping by land cover”.

References:

DC Morton, Y Le Page, R DeFries, GJ Collatz, GC Hurtt. Understorey fire frequency and the fate of burned forests in southern Amazonia. Philosophical Transactions of the Royal Society of the Royal Society of London B: Biological Sciences,368 (1619) 2013

31. Lastly the authors would do well to know better the deforestation and land cover change literature for the Amazon. The analysis does not seem to include, or reflect, the importance of areas of forest that may be degraded by selective logging. Although they recognize that fires could occur in secondary forests, they omit a discussion of degraded forest, which actually may be more important. This is important because there has been some recent debate over the coupling of selective logging and fire. The first studies, such as Cochrane et al (1999, 2001) argued that selective logging changes local moisture conditions, making logged forest susceptible to drying and then fire. In this model, the fire source would be a proximate deforestation event or pasture fire which would “escape” to completely or significantly burn dry, degraded logged forest. This model, however, was based on studies over limited geographic scale, generally a few

satellite images in a region around Paragominas.

When the entire Amazon was analyzed by Matricardi et al. (2013) the coincidence of forest degradation by logging and subsequent fire did not exist. Most fires in forests did not occur on land that was degraded by logging. Most fires were found in forest edges and intact forests. The interesting and important question here is whether increased drought and drought intensity could be catalyzing a change in this decoupled trend observed by Matricardi in contrast to the coupled condition postulated by Cochrane. There could be a trend now being observed by these authors toward much more fire in these degraded forests that are now increasingly dry and susceptible. Hence, while Matricardi observed only few cases where fire occurred following degradation by logging, climate and drought are changing conditions significantly. The authors need to consider this potential because it would suggest a shift in the recent patterns observed by Matricardi et al (2013) toward the model that had been locally idiosyncratic proposed by Cochrane et al (1999, 2001), and provide a precise mechanism to explain the dynamics of a situation where there is less outright deforestation yet increasing forest fire. This may be difficult since the PRODES program does not map forest degradation, but also be very interesting because it would suggest the increase in fire is indeed related to land use change – but greatly enhanced by climate change.

Answer 31: Thank you for the comprehensive comment. We completely agree with the reviewer. We mention in several parts of the paper the case of degraded forests, but do not give enough details. The investigation of the effect of drought-related fires on forests and the potentially catalyzing effect of selective logged forests is indeed very important, but we did not have the means to test it, as this would involved a completely independent analysis, using high resolution satellite data, that we feel would not reinforce our conclusions in the scope of this paper.

However, considering the relevance of this issue on the process, though, we have included a new paragraph to describe new figure 8 in the Discussion section that considers this effect. Figure 8 presents a conceptual model of the processes and feedbacks that lead to fire intensification in Amazonia. We have a key part of the paragraph that states:

“... (3) These factors increase the probability of fires to leak from open areas into adjacent human-modified forests (forest edges, fragments, burned, logged and secondary forests), which tend to be more susceptible to fire 39,40. Previous Amazonian-scale quantification (1992-1999) demonstrated that burned scars from fires were rarely detected in degraded logged forests 41, however, with the observed intensification of 21st century climate extremes and droughts a potential shift toward the previously postulated model 39,40 is expected. Based on this39,40, we suggest that in the 21st century climate degraded forests may become increasingly dry and susceptible to forest fires. (4) In comparison with intact forests, human-modified forests are characterized by greater canopy opening, larger stocks of dead material, drier microclimate, and lower wood density species 42 which are more susceptible to mortality during droughts 35. Such characteristics make these forests more flammable, facilitating fire incursion among the different forest types and also into (5) undisturbed old-growth forests.”

32. This study is completely reasonable in its assumptions and conclusions, and indeed I too would be prone to believe that climate change related drought would increase fire emissions even while outright deforestation rates are declining. The MOPITT CO data are important in this regard. I just want the authors to nail down the specifics and details in a more convincing and mechanistic way (e.g. explain how standing forests can burn and support their statement that fires “leak” to surrounding forests).

Answer 32: Thank you for the very positive and constructive review. Following recommendations from both reviewers 1 and 3 we have added a new figure (Figure 8) explaining the mechanisms of fire incidence and feedbacks between fire-climate-humans.

33. Some attention also needs to be given to the importance, or not, of increasing fire sizes vs increasing number of fires.

Answer 33: This is interesting but outside the scope of this paper. The conclusions of our paper are based on total area burned , so the shift in size would not affect the conclusions. However, we agree that the size distribution would allow a more in depth understanding of the process, providing insights into how fire is changing and what the main drivers are. Some of these issues were explored in a recent paper by Andela et al. (2017).

References:

N Andela, DC Morton, L Giglio, et al. A human-driven decline in global burned area. *Science* 356 (6345), 1356-1362 (2017).

34. Lastly there are many small technical and editing issues – the word data is plural, Extended data Fig 8 is missing, typos (e.g. MODIS product names).

Answer 34: Thank you. We have reviewed the text to increase accuracy.

REVIEWERS' COMMENTS:

Reviewer #1 (Remarks to the Author):

The revised manuscript, "21st Century drought-related fires counteract the decline of Amazon deforestation carbon emissions," is substantially improved. The authors adequately addressed all of the concerns I raised in my initial review. I particularly applaud the language clarification and new figure. I have no further concerns with this manuscript.

Reviewer #2 (Remarks to the Author):

The authors have thoroughly addressed all of my initial comments. Their willingness to perform an additional analysis (now Figure 7) has yielded some important insights that further bolster their original claims. I recommend this paper be accepted.

Reviewer #3 (Remarks to the Author):

This paper is much improved over the first submission and the authors were responsive to reviewers comments. I recommend that it be approved for publication. Having said this, I believe the authors continue to underestimate the uncertainty that arises from the use of coarse resolution data in mapping fires precisely on land cover. Lastly, I would like to note that in response 30 (reviewer #3) is an important point worth highlighting. It is also important to note that the ancillary issue -- ie whether degraded (logged) forest is subsequently burned is equally important and sorting out these kinds of synergies (and perhaps double counting) is important.

Responses to reviewers of the manuscript entitle:

“21st Century drought-related fires counteract the decline of Amazon deforestation carbon emissions”

By Aragão L.E.O.C. and co-authors

We thank all three Reviewers for their contributions towards the improvement of the final version of the manuscript. We feel that the final version of the manuscript presents much stronger evidences about our case and also is clearer for Nature Communications readers.

We are glad that all three Reviewers like the study and have recommended the acceptance of the paper for publication in Nature Communications.

Below, we repeat all Reviewers' comments and reply to the concerns one by one. Each comment is numbered, using a continuous sequence and with our responses in bold.

Reviewer #1

1. The revised manuscript, “21st Century drought-related fires counteract the decline of Amazon deforestation carbon emissions,” is substantially improved. The authors adequately addressed all of the concerns I raised in my initial review. I particularly applaud the language clarification and new figure. *I have no further concerns with this manuscript.*

Answer 1: Thank you for the relevant comments. We are sure that the revisions based on the reviewer's comments in the previous version of this paper have strengthened our narrative, results and conclusions.

Reviewer #2

2. The authors have thoroughly addressed all of my initial comments. Their willingness to perform an additional analysis (now Figure 7) has yielded some important insights that further bolster their original claims. *I recommend this paper be accepted.*

Answer 2: Thank you for the relevant comments. The suggested analysis for the previous version of the paper was critical for validating our claims. Conclusions are now much stronger. We really appreciate your contribution for the improvement of the paper.

Reviewer #3

3. This paper is much improved over the first submission and the authors were responsive to reviewers' comments. *I recommend that it be approved for publication.*

Answer 3: We thank the reviewer for the positive feedback and the careful revision of the manuscript. In this final version we have incorporated new text in the methods to mitigate the remaining concerns.

Reviewer 3 comment is listed below with number 4, followed by our reply.

4. Having said this, I believe the authors continue to underestimate the uncertainty that arises from the use of coarse resolution data in mapping fires precisely on land cover. Lastly, I would like to note that in response 30 (reviewer #3) is an important point worth highlighting. It is also important to note that the ancillary issue -- ie whether degraded (logged) forest is subsequently burned is equally important and sorting out these kinds of synergies (and perhaps double counting) is important.

In this final version we have highlighted in the Methods all points made by the reviewer. The reviewer's suggestion about highlighting in the text the points considered in response 30 (reviewer #3) of our previous revision was considered in different sections of the method. The reviewer was mainly concerned about making clear in the text the possible sources of uncertainties. So, we improved the text to explicitly present justifications for the issues pointed by the reviewer.

First, we added in the subsection "Burned area mapping by land cover" the sentence "Our burnt area estimate is likely to be conservative as the use of MODIS data at 250 m spatial resolution can underestimate the area burned by approximately 25% in relation to manually digitized burn scars based on 30m spatial resolution Landsat images^{52, 53}."

Second, we added to the subsection "CO₂ emissions from deforestation and forest fires" the following explanations:

- a) 'For this parameterization we did not considered the biomass from selectively logged forests that may have burned. Aboveground carbon in logged forests are estimated to be 35% lower than in undisturbed forests⁵⁹. However, the mean biomass value used in our study⁵⁸ includes selectively logged areas from 2000-2004, minimizing the impact of potential biomass overestimation in the calculation of CO₂ emissions. Moreover, the underestimation of burned area by our method, described above, can further counterbalance the overall biomass overestimation effect in the final CO₂ emissions estimates from forest fires.'
- b) 'We expect that the risk of double counting deforestation emissions as forest fire emissions is negligible, because just minimum fractions of the forest area that have burned are later deforested. Results from a previous analysis⁶⁰ demonstrated that only 2.6% of all burned forests between 1999 and 2008 were deforested by 2010.'